# Unleashing phosphorus mononitride

Simon Edin [1], Christian Sandoval-Pauker [2], Nathan J. Yutronkie [3], Zoltan Takacs[1], Fabrice Wilhelm [3], Andrei Rogalev [3], Balazs Pinter[2,5], Kasper S. Pedersen [4] ✉ & Anders Reinholdt [1] ✉

The interstellar diatomic molecule, phosphorus mononitride (P≡N), is highly unstable under conditions typical on Earth, and its utility for constructing elusive P–N π-bonded motifs has remained uncertain. Here, we show how Na(OCP) transfers a P atom to an electrophilic osmium nitride complex to form a metal-bound P≡N ligand. Quantum chemical calculations and X-ray absorption spectroscopy unveil a cumulenic [$Os^{IV}$=N=P] electronic structure comprising orthogonal Os=N and N=P π-bonding. On reaction with elemental sulfur, the highly reduced P≡N ligand, formally [PN]$^{2-}$, forms a trigonal planar [NPS$_2$]$^{2-}$ motif. Chlorination instead transforms the P≡N ligand to a bent [NPCl]$^-$ group coordinated to $Os^{III}$ ($S = \frac{1}{2}$). [3 + 2] cycloaddition of this radical with azide forms an aromatic interpnictide, [PN$_4$]$^-$, that is inaccessible from the parent P≡N system. These findings provide a rare glimpse of the divergent reactivity of the alien P≡N molecule, paving the way to long-sought P–N multiple-bonded archetypes.

Phosphorus mononitride (P≡N) was the first phosphorus-containing molecule to be identified in the interstellar medium, following Turner, Bally, and Ziurys's observations of its rotational lines from the Orion KL Nebula in 1987[1,2]. In our Solar System, P≡N has been detected in out-gassing vapor from a comet, suggesting the diatomic molecule may have been a prebiotic source of phosphorus on the early Earth[3]. The first example of man-made P≡N was reported by Herzberg in 1933; an electric discharge through a tube containing $N_2$ and $P_4$ generated the diatomic molecule in the gas phase, as authenticated from 24 rotational bands[4]. Given its vulnerable triple bond[5], studies of P≡N in condensed phase are far from trivial. Examples of synthetic strategies involve thermal decomposition of $P_3N_5$ at 800–900 °C followed by isolation in solid Kr[6], dehalogenation of [$N_3P_3Cl_6$] with Ag at 1300 K followed by isolation in solid Ar[5], or photolysis of [{1,2-$C_6H_4O_2$}P($N_3$)] in solid Ar[7]. As for reactivity studies, the radical derivatives PNH· and NPH· have been been studied in solid $N_2$[8], but even in cryogenic noble gas matrices, parent P≡N is unstable when the temperature rises above 10 K, which leads to oligomerization into discrete [$P_3N_3$] species[5,9,10], followed by the onset of higher polymerization products. To date, the smallest [$P_xN_y$] molecule isolated under ambient conditions is Klapötke's azido-phosphazene, [$P_3(\mu$-N)$_3$($N_3$)$_6$][11]; a smaller system such as P($N_3$)$_3$ decomposes rapidly in solution[12].

An attractive strategy for studying P≡N is to form an adduct that stabilizes this reactive functionality. Thus, in 1988, Niecke and co-workers reported aryl cations [Ar–N≡P]$^+$ (**A** in Fig. 1)[13,14], which display P≡N bond distances (1.475(8)–1.493(12) Å) similar to free, gaseous P≡N (1.49086(2) Å)[15]. Later, research teams led by Bertrand[16], Cummins[17], and Schulz[18] demonstrated how P≡N could be sandwiched between carbenes (**B**), anthracene (**C**), or cyclobutadienes (**D**). In view of the long P–N bonds (>1.69 Å) in these systems, along with the carbene C=P and C=N π-character in **B**, as well as the three-coordinate nature of the P and N atoms in **C** and **D**, these neutral species conform more closely to a description as having P–N single-bonds than P≡N triple bonds. Given that P≡N is valence-shell isoelectronic to $N_2$, the heterodiatomic molecule could serve as a π-backbonding ligand in a transition metal complex. Initial studies by Cummins and co-workers revealed low stability of the transient complex [{Ar[$^t$Bu]N}$_3$V–N≡P], which oligomerizes to diphosphene and *cyclo*-triphosphane derivatives (Ar = 3,5-Me$_2$C$_6$H$_3$)[19]. In 2020, Smith and co-workers isolated the first transition metal P≡N complexes by reductively coupling $Fe^{IV}$ nitride and $Mo^{VI}$ phosphide precursors to afford a heterobimetallic system, [Mo–PN–Fe]. Subsequent conversion of this complex with $^t$BuNC yielded a P≡N complex anion, [{N(CH$_2$CH$_2$NSiMe$_3$)$_3$}Mo–P≡N]$^-$ (**E**) with

[1]Department of Chemistry, Centre for Analysis and Synthesis, Lund University, 22100 Lund, Sweden. [2]Department of Chemistry and Biochemistry, University of Texas at El Paso, El Paso, TX 79968, USA. [3]ESRF-The European Synchrotron Radiation Facility, CS 40220, 38043 Grenoble, Cedex 9, France. [4]Department of Chemistry, Technical University of Denmark, Kemitorvet 207, DK-2800 Kgs, Lyngby, Denmark. [5]Present address: European Research Council Executive Agency, Brussels, Belgium. ✉e-mail: kastp@kemi.dtu.dk; anders.reinholdt@chem.lu.se

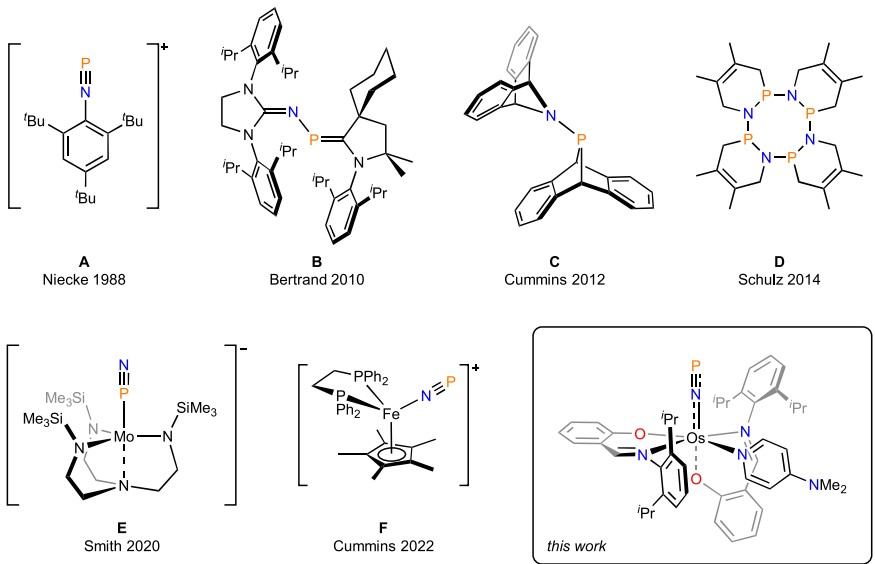

**Fig. 1 | Representative studies of P≡N-containing molecules and metal complexes.** **A** P≡N stabilized as a cationic aryl system. **B** P≡N trapped between two N-heterocyclic carbenes. **C** P≡N sandwiched between two anthracene units. **D** A cyclic tetramer of P≡N stabilized by butadiene units. **E** P≡N coordinated to molybdenum; photolysis converts this complex to its N-bound linkage isomer. **F** P≡N released from an anthracene-scaffolded azidophosphine and coordinated to an iron complex.

an iron-isonitrile complex as counter ion; in the solid state, **E** converts photolytically to its N-bound linkage isomer[20]. More recently, Cummins and co-workers thermolyzed an anthracene-scaffolded azidophosphine, [{C₁₄H₁₀}P(N₃)], to form anthracene, N₂, and P≡N; the heterodiatomic molecule was intercepted by a Fe$^{II}$ complex (**F**)[21]. Interestingly, the N-bonded linkage isomer of **F** is energetically preferred over P-bonded and $\eta^2$-coordination isomers by 14.3 and 16.8 kcal mol$^{-1}$, respectively.

The advanced synthetic protocols devised for making P≡N complexes have limited the utility of this functionality for constructing synthetically demanding phosphorus-nitrogen multiple bonded architectures by redox, atom transfer, or cycloaddition strategies. The only reported reactions of a P≡N ligand involve complex **E**, which can be metallated by a Rh$^{I}$ center or silylated by Me₃SiCl[20]. These conversions bear a striking resemblance to metallation and silylation of transition metal dinitrogen complexes[22], which calls into question whether N≡N and P≡N ligands display essentially identical reactivity. Given the elusive nature and uncharted reactivity of P≡N complexes, we sought alternative synthetic methodology toward this functionality and its chemistry. Herein, we describe how sodium phosphaethynolate delivers a P atom[23–36] to an electrophilic osmium nitride complex to form a neutral [Os−N≡P] ↔ [Os=N=P] motif. We probe its electronic structure by isotopic labeling, multinuclear NMR, vibrational spectroscopy, X-ray absorption near edge structure (XANES) spectroscopy, and theoretical studies. We also report how the P≡N ligand can be elaborated into unique inorganic motifs such as a trigonal planar [NPS₂]²⁻ and a bent [NPCl]⁻ group. The P≡N ligand does not react directly with azides to form an aromatic heterocycle, [PN₄]⁻, but is activated toward such [3 + 2] cycloaddition when subjected to umpolung.

## Results and discussion

### P-atom transfer to an osmium nitride to form P≡N complex 3

In spite of the high electronegativity of nitrogen (3.04, Pauling scale), high-valent, late transition metal nitride complexes possess such low d-orbital energies that their M≡N bonds may become polarized toward the metal center rather than the nitride ligand[37]. Given that group 8 nitrides react with elemental sulfur to form thionitrosyl complexes, [M−N≡S][38–40], and considering the isoelectronic relationship between S

and P⁻, we inquired whether a P≡N ligand could be assembled from a terminal osmium nitride functionality and a phosphorus atom transfer reagent such as Na(OCP)[23–36] To this end, we treated nitride complex (Bu₄N)[Os(N)Cl₄] with salicylaldimine ligand Na[salNdipp] in THF to form [(salNdipp)₂(Cl)Os≡N] (**1**, Fig. 2, sal = salicylidene, dipp = 2,6-diisopropylphenyl). Further conversion of **1** with AgOTf afforded triflate complex [(salNdipp)₂(OTf)Os≡N] (**2**) as orange crystals in 95% isolated yield after removal of AgCl. Treatment of **2** with Na(OCP) · 2.5 dioxane in THF led to several strong IR absorptions (1897–2202 cm$^{-1}$, suggesting coordinated CO ligand) as well as two $^{31}$P NMR singlets (233, 220 ppm). To obtain a cleaner conversion, we first blocked one coordination site on the osmium center by treating **2** with 4-dimethylaminopyridine (DMAP) and subsequently with Na(OCP) · 2.5 dioxane in THF. This led to effervescence over 5 min and a color change from light to dark orange. Very dark (almost black) crystals of [(salNdipp)₂(DMAP)Os(NP)] (**3**) were isolated in 94% yield after removal of NaOTf. X-ray crystallography revealed salNdipp⁻ ligands arranged with the imine nitrogen atoms in a *trans*-configuration, the phenolate oxygen atoms *cis*, and DMAP coordinated *cis* to the P≡N ligand (Fig. 3A). The phosphorus mononitride ligand displays linear coordination through nitrogen and a short P≡N bond, 1.536(5) Å, (elongated 3.0% relative to free P≡N)[15]. Remarkably, the Os−NP bond, 1.846(5) Å, is shorter than any Os−N₂ bond in a dinitrogen complex (shortest: 1.896(6) Å)[41]. As a direct comparison to **3**, we independently synthesized the N₂ analog, [(salNdipp)₂(DMAP)Os(N₂)] (**7**), vide infra. The Os−N₂ bond in **7**, 1.903(6) Å, is substantially longer than the Os−NP bond in **3**, which suggests P≡N to be a stronger π-acid than its homoatomic analog.

### Spectroscopic characterization of 3

To assess the electronic structure of **3** relative to P≡N complexes **E** and **F** (Fig. 1), we turned to spectroscopic methods (Fig. 3B–E). $^{31}$P NMR data evince a more shielded phosphorus nucleus in **3** (249 ppm) than in **E** (312 ppm) and **F** (272 ppm). Similarly, the isotopolog [(salNdipp)₂(DMAP)Os($^{15}$NP)] (**3-$^{15}$N**), displays a more shielded $^{15}$N NMR environment (d, 396 ppm, $^1J_{NP}$ = 62 Hz) than do **E** (445 ppm) and **F** (450 ppm). Whereas free P≡N displays an IR stretching frequency at 1323 cm$^{-1}$, complexes **3/3-$^{15}$N** display redshifted resonances at 1258/1221 cm$^{-1}$, in line with weakening of the P≡N bond upon complexation[6,42]. By contrast, **F** displays a higher-

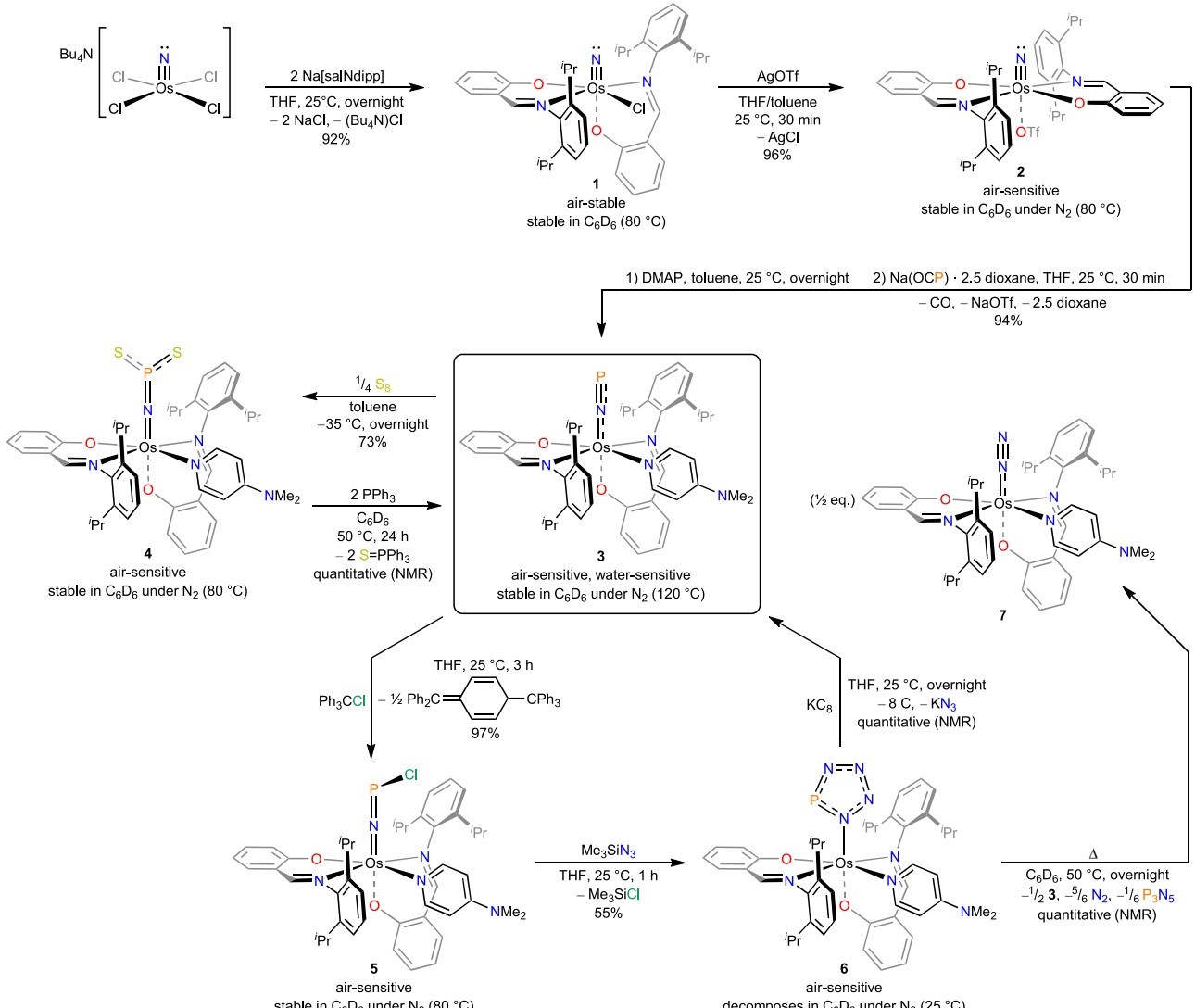

**Fig. 2 | Synthesis and reactivity of a P≡N complex (3).** Conversion of (Bu₄N)[Os(N) Cl₄] to nitride precursors **1** and **2**. Treatment of **2** with DMAP and Na(OCP) yields P≡N complex **3**. Oxidation of **3** with S₈ and Ph₃CCl generates [NPS₂]²⁻ complex **4** and [NPCl]⁻ complex **5**, respectively. Complex **5** reacts with Me₃SiN₃ to form [PN₄]⁻ complex **6**, which decomposes thermally to **3** and N₂ complex **7**.

energy vibration (¹⁴N/¹⁵N: 1271/1238 cm⁻¹), suggesting more limited π-backbonding in this iron complex. Moreover, UV–vis spectroscopy revealed intense absorptions at 258, 318, 348, and 401 nm (THF, $\varepsilon$ = 45,000, 26,000, 27,000, and 18,000 M⁻¹ cm⁻¹, respectively), indicative of charge-transfer transitions between the Os center and the P≡N ligand. Overall, these spectroscopic observations suggest the presence of more extensive π-backbonding in the π-basic Osᴵᴵ system, **3**, than observed in the Moᴵᴵ and Feᴵᴵ complexes, **E** and **F**.

### Electronic structure of 3

We further studied the electronic structure of **3** by density functional theory (DFT), using PBE0/def2-TZVP(-f) for geometry optimizations and TPSSh/def2-TZVP for single-point calculations. In theory, **3** can be described by three limiting resonance contributors, namely [Osᴵᴵ] singly bonded to a neutral [P≡N] ligand, [Osᴵⱽ] doubly bonded to a dianionic [P=N]²⁻ ligand, or [Osⱽᴵ] triply bonded to a tetraanionic [P–N]⁴⁻ ligand. Upon reaction with a hypothetical electrophile, these scenarios could lead to single (E₁), twofold (E₂), or threefold (E₃) functionalization of the P≡N ligand (Fig. 4A). In a molecular orbital depiction, an [OsNP] fragment possesses 5d$_{xz}$, 5d$_{yz}$, 2p$_x$, 2p$_y$, 3p$_x$, and 3p$_y$ atomic orbitals aligned for π-bonding (Fig. 4B), yielding linear combinations that are in-phase

bonding (π), non-bonding with a nodal plane at N (π$_{nb}$), and out-of-phase antibonding (π*). Calculated molecular orbitals for **3** (Fig. 4C) unambiguously show the two π-systems corresponding to the in-phase bonding π set; however, the unsymmetrical ligand field around osmium concentrates HOMO − 23 at N–P, whereas HOMO − 20 is Os–N centered. These localized and orthogonal N=P and Os=N π-bonds suggest that **3** possesses a dominant cumulenic character, [Os=N=P]; strong π-backdonation from osmium confers an [N=P]²⁻ character on the diatomic ligand. Looking to higher energy, the π$_{nb}$ orbitals (HOMO − 2, HOMO) have large amplitudes on Os and P, with a nodal plane on the central N atom. Interpreted as Lewis structures, the π$_{nb}$ orbitals represent two d-electrons on Os and one lone pair on P. Concomitant with this notion, electrophilic Fukui functions display large amplitudes on Os and P (Supplementary Fig. 47), pointing out likely loci of electrophilic functionalization (vide infra). Finally, the out-of-phase π* set is, as expected, energetically low-lying virtual orbitals (LUMO + 2 and LUMO + 3).

In harmony with the topology of the π and π$_{nb}$ molecular orbitals, Mayer bond orders (Fig. 4D) also indicate multiple bond character for both the N–P bond (2.22) and the Os–N bond (1.21), suggesting a dominant [Os=N=P] Lewis structure. This electronic structure was

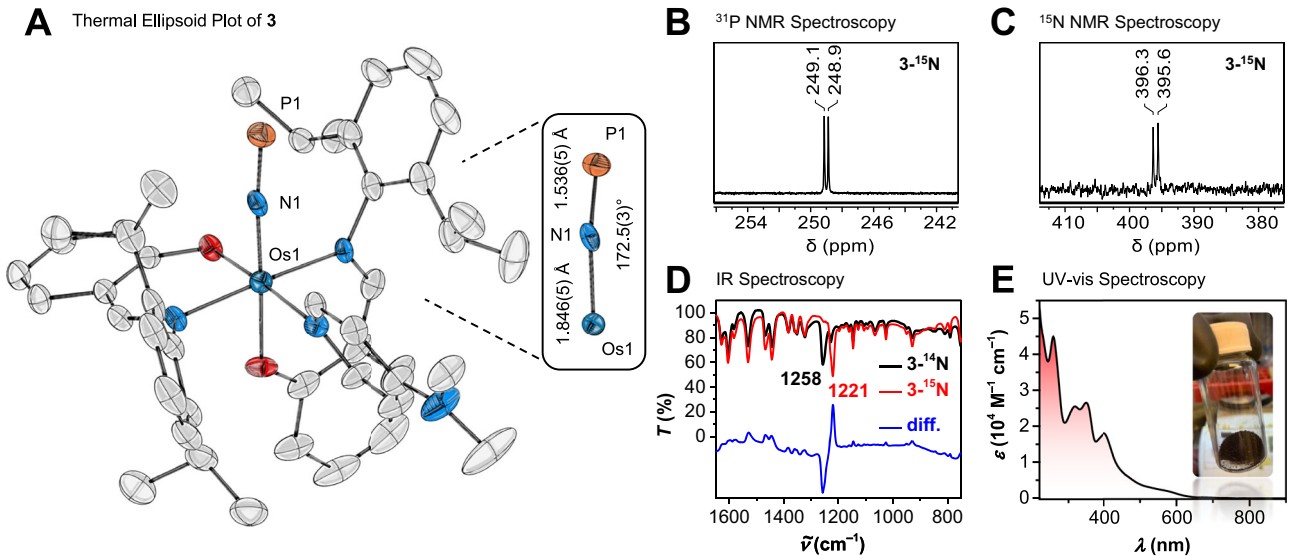

**Fig. 3 | Structural and spectroscopic studies of 3. A** Molecular structure of P≡N complex **3** (50% probability, 100(2) K; H atoms and toluene omitted). **B** $^{31}$P NMR data (**3-**$^{15}$**N**, 162 MHz, C$_6$D$_6$). **C** $^{15}$N NMR data (**3-**$^{15}$**N**, 81 MHz, C$_6$D$_6$). **D** IR data for **3-**$^{14}$**N** (black trace) **3-**$^{15}$**N** (red trace), and difference spectrum (blue trace). **E** UV−vis data (2.3 × 10$^{-5}$ M **3** in THF); inset shows crystals of **3**.

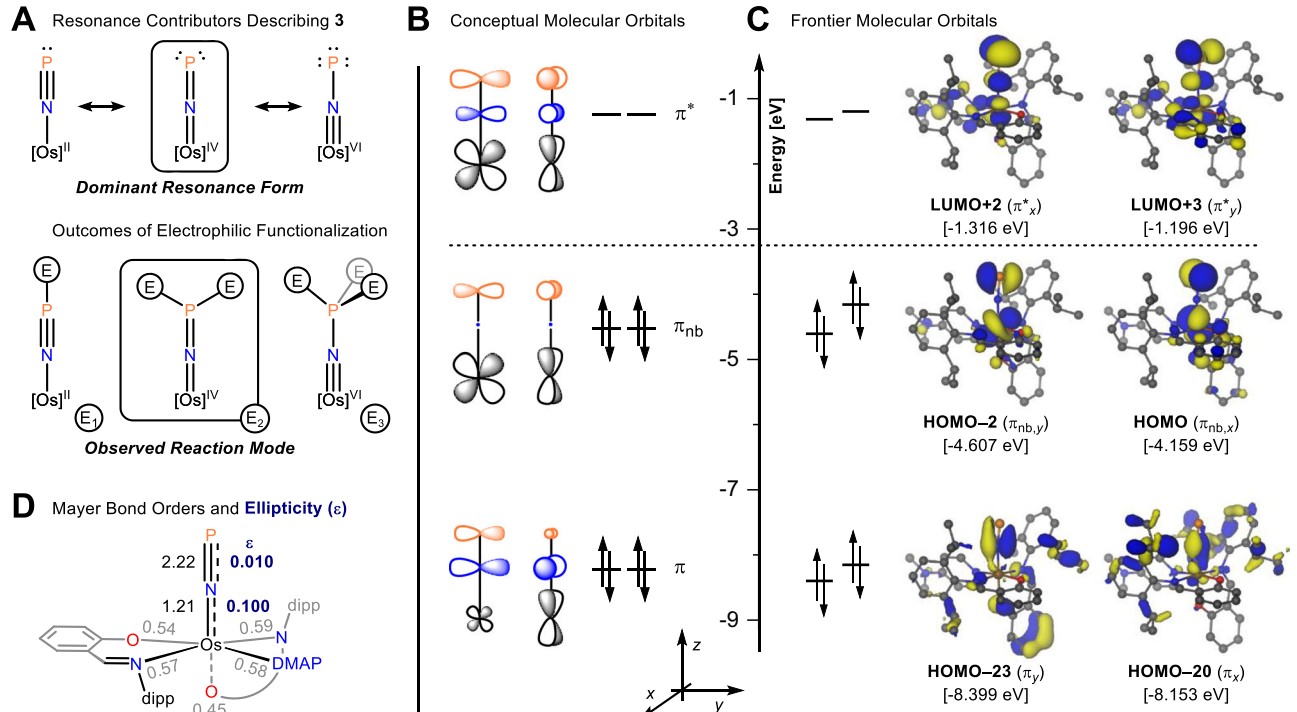

**Fig. 4 | Electronic structure studies of 3. A** Possible resonance contributors for **3** along with products expected from electrophilic functionalization of the P≡N ligand. **B** Conceptual π molecular orbitals formally describing the [Os=N=P] bonding pattern of **3**. **C** Frontier molecular orbitals of **3** showing the localized and orthogonal Os=N and N=P π-type interactions along the cumulenic [Os=N=P] functionality (HOMO − 23, HOMO − 20), calculated using DFT (TPSSh-D3/def2-TZVP, plotted at an isovalue of ±0.04 a.u.). **D** Mayer bond orders and bond ellipticities (in bold, purple) for **3**.

further corroborated using Wiberg bond orders (Supplementary Table 6) and a Natural Bond Orbital (NBO) analysis (Supplementary Tables 7 and 8). Topological analysis using quantum theory of atoms in molecules (QTAIM) reveals moderate ellipticity ($\varepsilon = 0.100$) for the Os=N bond, supporting a non-symmetric electron distribution within this π-bond, whereas the N=P bond is closer to cylindrical symmetry ($\varepsilon = 0.010$). When comparing the electronic structure of **3** to the few reported P≡N systems (Fig. 1), both **E** and **F** display very marginal π-backbonding and are best represented by P≡N triple bonds[20,21]. In line

with this bonding scheme, Smith showcased how silylation and metallation led to linear [Mo−P≡N−X] motifs (X = SiMe$_3$, Rh$^I$), indicating E$_1$ reactivity in Fig. 4A. Conforming more closely to an [Os=N=P] description, **3** could conceivably undergo twofold electrophilic functionalization in line with E$_2$ reactivity.

## Oxidation of 3 by sulfur to form an [NPS$_2$]$^{2-}$ ligand

To probe the reactivity of the osmium-coordinated P≡N ligand, we treated **3** with elemental sulfur. Considering Fig. 4A, this could

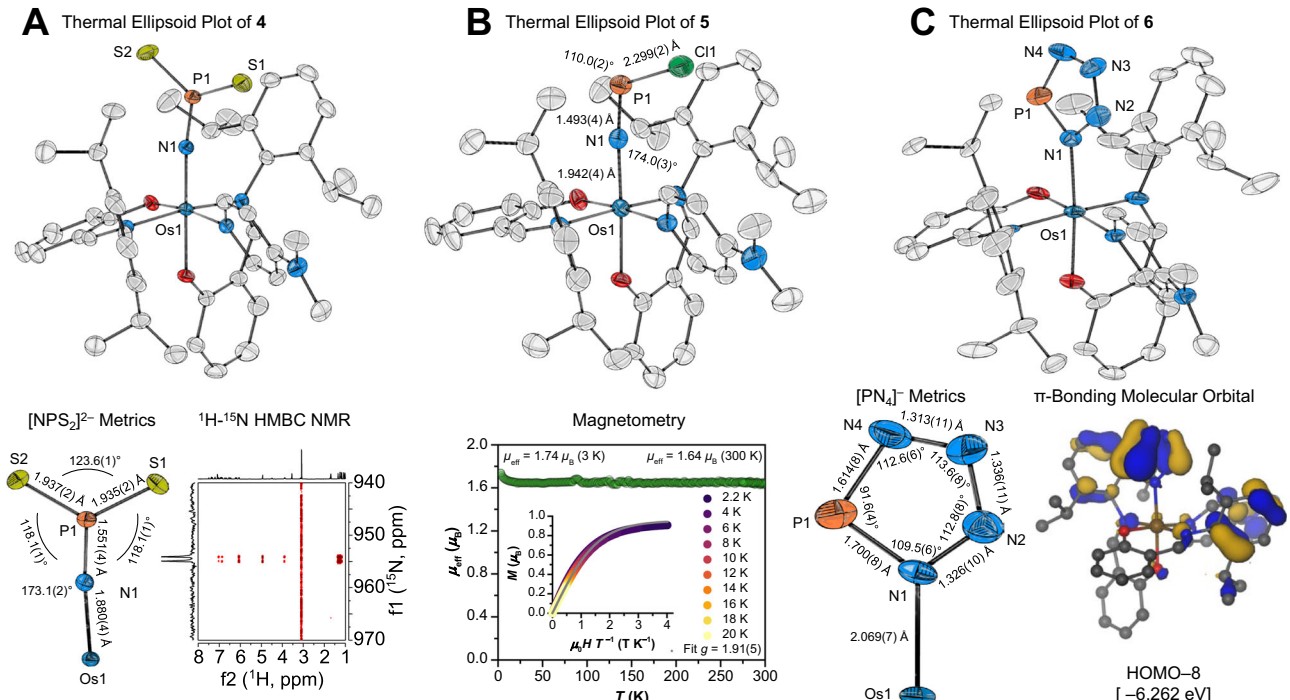

**Fig. 5 | Structural studies and characteristic data of products derived from 3. A:** X-ray structure of **4** (50% probability, 100(2) K; H atoms, toluene omitted), along with metric and $^1H-^{15}N$ HMBC data for the $[NPS_2]^{2-}$ fragment. **B** X-ray structure of **5** (50% probability, 100(2) K; H atoms, hexane omitted) along with magnetic studies showing the effective magnetic moment of the complex ($\mu_{eff}$, green circles) at 3–300 K and the magnetic field dependence of the magnetization, $M$ versus $\mu_0HT^{-1}$ (inset; colored dots, including a fit to the $S = \frac{1}{2}$ Brillouin function shown as gray trace). **C** X-ray structure of **6** (50% probability, 100(2) K; H atoms, hexane omitted) along with metrics within the $[PN_4]^-$ fragment as well as one of the π-bonding molecular orbitals of **4**, illustrating the aromatic character of the interpnictide ring.

generate ternary ions such as $[NPS_2]^{2-}$ or $[NPS_3]^{4-}$ or a heavy-atom analog of nitrous oxide, NPS[43]. When using ¼ eq. $S_8$, diamagnetic $[(salNdipp)_2(DMAP)Os(NPS_2)]$ (**4**) formed in 73% yield as dark orange crystals. The conversion is well conducted in toluene, which dissolves **3** and $S_8$, whereas the more sparingly soluble oxidation product, **4**, crystallizes directly from the reaction mixture. Desulfurization of **4** with $PPh_3$ regenerated **3** along with $Ph_3PS$, showcasing mild and chemically reversible formation of the P–S bonds. X-ray crystallography revealed an osmium center bound to a trigonal planar nitridodisulfidophosphate(V) ligand, $[NPS_2]^{2-}$, coordinated linearly through N (Fig. 5A). The planar $P^V$ center (angle sum 359.8°) contrasts with the pyramidalized $P^{III}$ center (311.9°) in the known complex, $[\{Ar[^tBu]N\}_3V-\{NPS_2(C_{10}H_6)\}]$, formed by trapping transient $[\{Ar[^tBu]N\}_3V^{III}-N≡P]$ with 1,8-naphthalenediyl disulfide $(C_{10}H_6)S_2$[19]. The Os–N, P–N, and P–S bond distances in **4** indicate π-delocalization in the $[OsNPS_2]$ fragment, which is corroborated by Mayer bond orders (1.30–1.67). The formation of **4** upon $E_2$-type reactivity (Fig. 4A) demonstrates the striking contrast in reactivity of **3** compared to the few P≡N complexes reported to date. Spectroscopically, **4** displays a $^{31}P$ NMR singlet at a lower chemical shift than **3** (233 ppm). A $^1H-^{15}N$ HMBC experiment revealed eight long-range correlations ($^4J_{HN}$, $^6J_{HN}$, $^7J_{HN}$) to the $[NPS_2]^{2-}$ ligand, unequivocally identifying a low-intensity and strongly deshielded $^{15}N$ doublet at 955 ppm ($^1J_{NP} = 51$ Hz, Fig. 5A).

### Chlorination of 3 to form an [NPCl]⁻ ligand

To test for radical reactivity, we treated **3** with a mild chlorinating agent such as trityl chloride. This generated Gomberg's dimer (observed by $^1H$ NMR, and readily separated by washing with hexane) along with a mono-chlorinated $Os^{III}$ product $[(salNdipp)_2(DMAP)Os(NPCl)]$ (**5**), isolated as dark orange crystals in 97% yield. Excess trityl chloride did not perturb the product distribution, showcasing how this choice of oxidant prevents overoxidation. The $[NPCl]^-$ ligand is the isoelectronic phosphorus analog of the rare main group terminal nitride, thiazyl

chloride, $[NSCl]$[44]. X-ray crystallography revealed linear Os–N–P and bent N–P–Cl geometries, in line with a lone pair on phosphorus (Fig. 5B). Interestingly, **5** possesses one of the longest structurally characterized P–Cl bonds (top 1%)[45,46], suggesting a labile halogen substituent. Given its radical nature, **5** displays paramagnetically shifted $^1H$ NMR resonances from −22 to +23 ppm (FWHM = 5–1100 Hz). Magnetization measurements (Fig. 5B) revealed an effective magnetic moment ($\mu_{eff} = 1.64 \mu_B$, 300 K) in accord with the spin-only value for an effective $S = \frac{1}{2}$ system. The magnetic moment remains practically constant until 20 K but increases slightly at the lowest temperature to reach 1.74 $\mu_B$ at 3 K, suggesting the presence of weak ferromagnetic interactions between the constituent molecules. All magnetic field and low-temperature magnetization data collapse on a single curve when plotted against the reduced variable $\mu_0HT^{-1}$, which is well described by an effective $S = \frac{1}{2}$ Brillouin function with $g = 1.91(5)$ (Fig. 5B, inset). The significant reduction of $g$ from 2.0 suggests a dominant metallo-radical character of **5**. In accord, computed Löwdin spin densities (Supplementary Fig. 48) indicate an $Os^{III}$-centered radical with minor spin delocalization onto the nitrogen of $[NPCl]^-$.

### [3 + 2] Cycloaddition of 5 with azide

In spite of the hypothetical triple bond reactivity of a P≡N ligand, **3** does not undergo [3 + 2] cycloaddition with azides to form a $[PN_4]^-$ heterocycle. However, the oxidized species, **5**, offers a P≡N fragment subjected to umpolung, and upon reaction with $Me_3SiN_3$, a thermally sensitive tetrazaphospholide $Os^{III}$ complex, $[(salNdipp)_2(DMAP)Os(\eta^1-N_4P)]$ (**6**), forms as orange crystals in 55% yield. Aromatic pnictogen rings, and nitrogen-rich interpnictides in particular, are challenging constructs. Baudler and co-workers reported $[P_5]^-$ in 1987[47,48], but it was not until 2016 and 2017, that Velian and Cummins isolated the first unsubstituted $[P_2N_3]^-$ heterocycle[49], and Lu and co-workers isolated $[N_5]^-$, respectively[50,51]. All other unsubstituted $[P_nN_{5-n}]^-$ rings have remained unknown[52,53]. X-ray crystallography identified **6** as a single

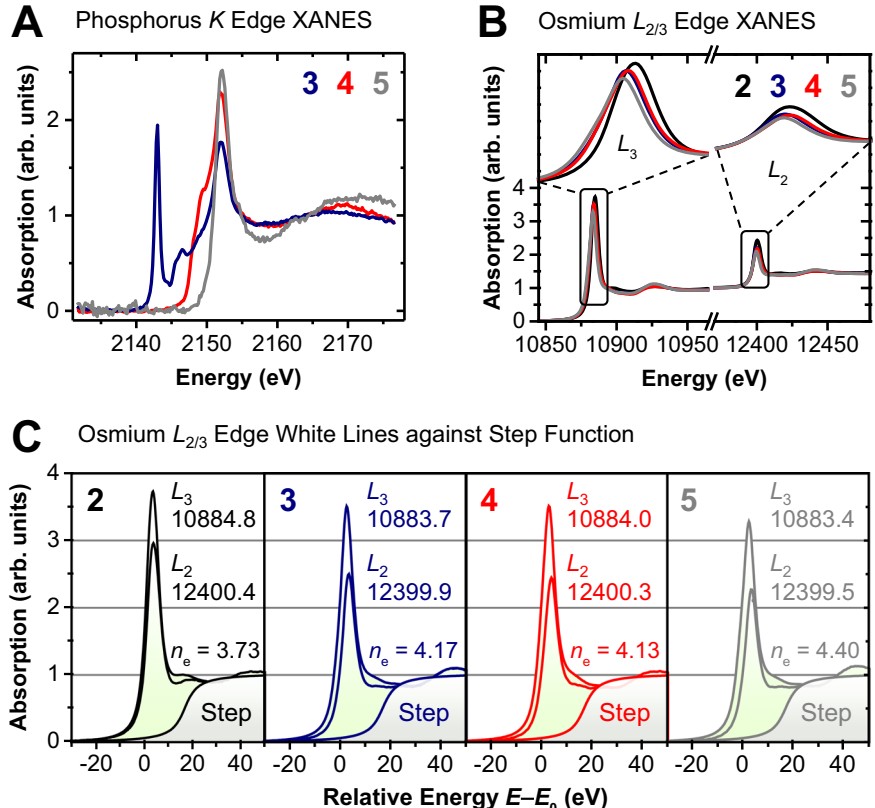

**Fig. 6 | Normalized XANES spectra. A** Phosphorus $K$-edge for **3** (blue trace), **4** (red trace), and **5** (gray trace). **B** Osmium $L_3$- and $L_2$-edges for **2** (black trace), **3** (blue trace), **4** (red trace), and **5** (gray trace). **C** Overlay of osmium $L_{2/3}$-edge XANES spectra for **2–5** within a common energy range ($E-E_0$) where $E_0$ corresponds to the photon energy of the $L_3$ white line maxima while the respective $L_2$ spectra were shifted in energy, such that the post-edge region superimposed with the $L_3$ spectra.

Values of the $E_0$ (in eV) for each spectrum are given below their respective edge label. The integrated intensities of the white lines used for the sum rule analysis were obtained by subtracting the step function that reflect transitions into continuum common for both $L_3$ and $L_2$-edges (shaded gray region). The sum rule analysis quantifies the number of electrons populating the $5d_{5/2}$ and $5d_{3/2}$ sub-levels ($n_e$).

linkage isomer with a planar $[PN_4]^-$ ring coordinated through a nitrogen adjacent to phosphorus (Fig. 5C). The N−N bonds are alike within 0.02 Å, while the two P−N bonds differ by almost 0.1 Å. Bond angles around the nitrogen atoms lie fairly close to the value for a regular pentagon, whereas the P−N bonds are practically orthogonal. The Os−N₄P bond distance is alike the other Os−N$_{imine/DMAP}$ bonds of **6**, implying coordination through a dative σ-bond. Computational analyses (bond indices, MO, NBO) affirm this description and also reveal delocalized π-bonding and aromatic character of the $[PN_4]^-$ ring (Fig. 5C), coordinated to an Os$^{III}$-centered radical (Supplementary Fig. 49). Being paramagnetic, **6** displays broad ¹H NMR resonances from −30 to +19 ppm (FWHM = 10–1500 Hz) and a magnetic moment of 1.85 μ$_B$ in solution (Evans' method, THF, 298 K).

When **6** is dissolved in $C_6D_6$, the complex thermally decomposes at room temperature over the course of days. This generates a 1:1 mixture of **3** and a dinitrogen complex, $[(salNdipp)_2(DMAP)Os(N_2)]$ (**7**), along with a white solid, attributable to binary phosphorus(V) nitrogen species[54]. The N₂ complex could be separated after selectively converting **3** into **4** with S₈. An IR analysis of the decomposition residue identified **3** (1258 cm⁻¹) and **7** (¹⁴N≡¹⁴N, 2021 cm⁻¹). On the other hand, **6-¹⁵N** converts to **3-¹⁵N** (1221 cm⁻¹) and **7-¹⁵N** (¹⁵N≡¹⁴N, 1988 cm⁻¹), without showing scrambling of the ¹⁵N label with ¹⁴N (Supplementary Fig. 35). The retention of the label suggests that this nitrogen remains coordinated to osmium throughout the decomposition process, that it is unlikely that the $[PN_4]^-$ ring slides in such a way that it attains $\eta^2$ or higher hapticities, and that **7** does not acquire its N₂ ligand from the atmosphere (experiments conducted under N₂). To gain more insight into the mode of decomposition, we modeled the thermodynamics of

unimolecular rupture of the $[PN_4]^-$ ring in **6** by DFT (Supplementary Table 19). Amongst the considered possibilities, the only thermodynamically feasible process corresponds to the formation of **7** and a $[PNN]^•$ radical ($\Delta G^\ominus_{(benzene)} = -2.34$ kcal mol⁻¹). This preliminary result implies that the selective cleavage of the $[PN_4]^-$ ring might provide access to the highly elusive $[PNN]^•$ radical, which was recently identified in a frozen Ar matrix at 10 K[55].

Finally, the metastable nature of **6** is also manifested in its reduction chemistry; treatment with KC₈ does not produce a hypothetical salt such as "$[K][(salNdipp)_2(DMAP)Os(\eta^1-N_4P)]$", but instead leads to a clean retro [3 + 2] cycloaddition, forming **3** and KN₃ (identified by ¹H/³¹P NMR and IR data, $\nu_{N_3^-}$ = 2012 cm⁻¹).

## X-ray spectroscopic studies

To gain element-specific information about the electronic structure of **2–5**, we turned to X-ray absorption near edge structure (XANES) spectroscopy. For the phosphorus $K$-edge (Fig. 6A), the dipole selection rule $\Delta l = \pm 1$ allows excitations from filled core 1s levels to vacant valence orbitals having 3p character. Complexes **3–5** display drastic spectral differences, which are brought about by a subtle interplay between the oxidation state of phosphorus, covalent orbital interactions with neighboring atoms, and the local geometry about the phosphorus atom[56]. These spectral features directly map the profound differences in electronic structure for the P≡N, $[NPS_2]^{2-}$, and $[NPCl]^-$ ligands. We also recorded osmium $L_{2,3}$ XANES to probe directly the electronic configuration of the 5d orbitals of osmium (Fig. 6B). The spectra show strong resonances (white lines) at the absorption edges, corresponding to dipole-allowed $2p_{1/2, 3/2} \rightarrow 5d_{3/2, 5/2}$ transitions. Of the

series of compounds, **2** features the highest white line photon energies, which are slightly higher than for both **3** and **4**. The lowest photon energies are observed for **5** commensurate with the lower, formal oxidation state assignment. In addition, the white line integrals are a sensitive measure of the number of electron holes in the Os 5d states (Fig. 6C), as previously reported[57,58]. By applying the spin-orbit sum rules for p → d transitions[59], the number of electrons populating the $5d_{5/2}$ and $5d_{3/2}$ sublevels ($n_e$) were quantitatively determined. This, in congruence with the energy positions of the white lines, confirms the highest oxidation state in **2** ($n_e = 3.73$; formally $Os^{VI}$), a common, intermediate oxidation state in **3** and **4** ($n_e = 4.17$ and 4.13; formally $Os^{IV}$), and the lowest oxidation state in **5** ($n_e = 4.40$; formally $Os^{III}$). Bear in mind that the present analysis does not provide the true oxidation state since any contributions arising from transitions to the 6s and 6p states are neglected. The spectroscopic findings can be well understood in terms of ligand-centered oxidations of a $[PN]^{2-}$ moiety, forming closed-shell motifs, $[NPS_2]^{2-}$, and $[NPCl]^{-}$, engaged in cumulenic bonding with osmium. Remarkably, as evidenced from XANES spectroscopy, oxidation of **3** to **4** leaves the oxidation state of osmium invariant, whereas oxidation of **3** to **5** leads to a lower oxidation state of osmium.

## Implications of this study of P≡N

Overall, the interstellar diatomic molecule, phosphorus mononitride (P≡N), offers singular access to phosphorus-nitrogen multiple bonded constructs, serving as exemplars in structure, bonding, and reactivity. Only a few synthetic strategies to stabilize P≡N have been reported. We have demonstrated how P-atom transfer from Na(OCP) to an osmium nitride assembles a terminally bound P≡N ligand (**3**). We envisage this strategy can be generalized to other nitride precursors, provided that they are electrophilic in nature, and thus give access to a wide array of transition metal P≡N complexes. For our osmium-based system (**3**), theoretical modeling coupled with XANES spectroscopy revealed a highly reduced P≡N functionality, formally $[PN]^{2-}$, and a cumulenic $[Os^{IV}=N=P]$ electronic structure with orthogonal Os=N and N=P π-bonding. The P≡N ligand, being both coordinatively stabilized and reductively activated by the π-basic osmium scaffold, undergoes oxidative conversions into unique binary and ternary π-bonded main-group motifs, including trigonal planar $[NPS_2]^{2-}$ (**4**), bent $[NPCl]^{-}$ (**5**), and cyclic $[NP_4]^{-}$ (**6**). In addition to the synthetic utility of P≡N complex **3** itself, we note the ability of **3**/**4** to form P–S bonds in a chemically reversible fashion, hinting at possible group transfer reactivity of the $[NPS_2]^{2-}$ ligand. Moreover, halide exchange of **5** could open the path to synthetically demanding $[NPX]^{-}$ fragments, heterocycles, and radical cations. We are presently pursuing these strategies toward coordinated intermediates that cannot be accessed through classic synthetic chemistry.

## Methods

The synthesis of complexes **1**–**7** were performed in Vigor glove boxes under a purified atmosphere of $N_2$ ($O_2 < 1$ ppm, $H_2O < 1$ ppm). Hexane and toluene were initially purified with an MBraun SPS system. Tetrahydrofuran and diethyl ether were initially stored over sodium benzophenone ketyl diradical, distilled by trap-to-trap transfer in vacuo, and degassed by freeze-pump-thaw cycles. Benzene-$d_6$ and THF-$d_8$ were stored over a potassium mirror overnight, sublimed/distilled by trap-to-trap transfer in vacuo, and degassed by freeze-pump-thaw cycles. The water content of the solvents was further reduced by storage over 4 Å molecular sieves. Celite and 4 Å molecular sieves were activated in vacuo overnight at 175 °C.

### Synthesis of $(Bu_4N)[Os(N)Cl_4]$

*Caution: $[OsO_4]$ is volatile, and exposure to its fumes may result in blindness.* Two reported procedures[60,61] were adapted as follows: A solution of KOH (1.0 g, 18 mmol, 4.53 eq.) in 1 ml water was prepared. In a well ventilated fumehood, an ampule containing $[OsO_4]$ (1.00 g, 3.93 mmol, 1.00 eq.) was opened, and the KOH solution quickly added to the ampule, resulting in a dark brown solution inside. Then, dropwise addition of an aqueous solution of $NH_3$ (6 M, 0.66 ml, 3.96 mmol, 1.00 eq.) was carried out until the point where the dark brown color was replaced by the golden color of $K[Os(N)O_3]$ crystals separating from the reaction mixture. The $[OsO_4]$ starting material tends to stick to the bottom of the ampule, and to ensure effective mixing of the reactants, the solution may be heated gently (40 °C) and agitated by pipette. (*Note: Excess $NH_3$ must be avoided due to the risk of precipitating $[NH_4][Os(N)O_3]$*). The procedure was repeated using a total of 5 ampules of $[OsO_4]$. The combined crystalline material was collected on a filter frit and washed quickly with $1 \times 5$ ml ice-cold water. These crystals were then dissolved in 15 ml boiling water and re-crystallized by cooling the solution to 0 °C overnight. The golden-yellow crystals were collected on a filter frit and dried in a dynamic vacuum. Yield of $K[Os(N)O_3]$: 5.43 g, 18.6 mmol, 94.8% based on $[OsO_4]$. In the next step, $K[Os(N)O_3]$ (5.43 g, 18.6 mmol) was dissolved in 150 ml water at 50 °C. A solution of $(Bu_4N)Cl$ in 25 ml $H_2O$ was added over 5 minutes, to rapidly produce a pale yellow precipitate, which was collected on a filter frit, washed with water ($2 \times 10$ ml), and dried in a dynamic vacuum over 1 h to yield $(Bu_4N)[Os(N)O_3]$ (9.11 g, 18.4 mmol, 98.8%, based on $K[Os(N)O_3]$). Next, the yellow powder of $(Bu_4N)[Os(N)O_3]$ was suspended in aqueous hydrochloric acid (4 M, 120 ml, 480 mmol), resulting in the formation of gaseous $Cl_2$ and a dark red suspension containing $(Bu_4N)[Os(N)Cl_4]$, which was left to stir overnight at 70 °C. The red precipitate was collected on a filter frit, washed with water ($3 \times 10$ ml), and dried in a dynamic vacuum over 1 h. Subsequently, $(Bu_4N)[Os(N)Cl_4]$ was recrystallized by dissolving in a minimal amount of $CH_2Cl_2$ (ca. 100 ml) and precipitating the complex as red crystals by slowly adding twice the volume of hexane to the solution. The red crystals were washed with hexane ($3 \times 10$ ml) and dried in vacuo. Yield of $(Bu_4N)[Os(N)Cl_4]$: 9.02 g, 15.3 mmol, 82.2% based on $K[Os(N)O_3]$. **IR**, solid ATR, $\nu$ (cm$^{-1}$): 1125/1090 for $(Bu_4N)[Os(N)Cl_4]$/$(Bu_4N)[Os(^{15}N)Cl_4]$.

### Synthesis of H(salNdipp)

The reported procedure[62] was adapted as follows: 2,6-diisopropylaniline (34.1 g, 90%, 173 mmol) and salicylaldehyde (25.8 g, 211 mmol) were mixed, resulting in a brown solution. Formic acid (0.1 ml) was added to catalyze the condensation reaction, leading first to a yellow oil containing water droplets, and subsequently a yellow solid. The yellow solid was recrystallized from hot methanol (ca. 250 ml), and the resulting yellow crystals were washed with cold methanol ($3 \times 10$ ml) and dried in vacuo. Yield of H(salNdipp): 43.8 g, 156 mmol, 89.9% based on 2,6-diisopropylaniline. **$^1$H NMR**, 400 MHz, CDCl$_3$, δ (ppm): 13.12 (s, 1H, OH), 8.34 (s, 1H, imine-CH), 7.45 (ddd, $J = 8.3, 7.3, 1.7$ Hz, 1H, aryl-CH), 7.39 (dd, $J = 7.7, 1.7$ Hz, 1H, aryl-CH), 7.22 (overlapped m, $J = 0.9$ Hz, 3H, aryl-CH), 7.10 (dd, $J = 8.3, 1.1$ Hz, 1H, aryl-CH), 7.00 (td, $J = 7.5, 1.1$ Hz, 1H, aryl-CH), 3.04 (hept, $J = 6.8$ Hz, 2H, $^i$Pr-CH), 1.22 (d, $J = 6.9$ Hz, 12H, $^i$Pr-CH$_3$).

### Synthesis of Na(OCP) · 2.5 dioxane

The reported procedure[23] was adapted as follows: In a 3-neck flask fitted with an $N_2$ inlet, 500 ml 1,2-dimethoxyethane (DME) was added and sparged with argon over 30 min. Sodium pieces (16.1 g, 700 mmol, 3.1 eq.), red phosphorus (7.0 g, 226 mmol, 1.0 eq.), and naphthalene (1.5 g, 12 mmol, 0.05 eq.) were added through one side-arm and left to stir at room temperature over three days, resulting in a color change from a red suspension to a black suspension with a green supernatant (due to the presence of sodium naphthalenide). The black suspension was cooled in an ice bath, and $^t$BuOH (sparged with $N_2$, but kept above 25 °C to avoid freezing) was added dropwise over 20 min, and the solution was allowed to warm to room temperature and stir for 1 h. Then, the solution was again cooled to 0 °C, and a solution of ethylene carbonate (20.0 g, 227 mmol, 1.0 eq.) in 50 ml DME (sparged with $N_2$)

was added dropwise over 1 h to give a greenish-brown suspension, which was left to stir overnight, gradually turning to a more yellow color. The solvent was removed under reduced pressure, and the solid residue was dissolved in 500 ml THF (sparged with $N_2$) and filtered through a glass frit. To this solution, 200 ml dioxane (sparged with $N_2$) was added to precipitate Na(OCP) · 2.5 dioxane as a cream-colored crystalline material, which was collected on a filter frit, dried in vacuo, and taken to the glove box. Yield of crude Na(OCP) · 2.5 dioxane: 18 g, 60 mmol, 26% based on P. Before being used to synthesize complex **3**, Na(OCP) · 2.5 dioxane was recrystallized by dissolving the salt in strictly dry THF in the glovebox and precipitating using anhydrous dioxane. **[1]H NMR**, 400 MHz, THF-$d_8$, δ (ppm): 3.56 (s, 8H, dioxane). **[31]P{[1]H} NMR**, 162 MHz, THF-$d_8$, δ (ppm): −394.30.

### Synthesis of $KC_8$

The reported procedure[63] was adapted as follows: Inside the glovebox, a thickwalled high-pressure reaction vessel was charged with a piece of potassium (1.16 g, 29.7 mmol, 1.00 eq.), graphite flakes (2.85 g, 237 mmol, 8.00 eq.), and a glass-coated stirbar. To make an optimally air-tight seal, the teflon screwcap of the reaction vessel was lined with teflon tape from the inside, and secured with electrical tape from the outside. Then, the reaction vessel was taken out of the glovebox and placed in an oilbath (150 °C). The color of the graphite flakes turned from black to golden ($KC_8$) within 5 min, and the reaction mixture was kept stirring at 150 °C over 2 h. The reaction vessel was taken back into the glovebox, and the golden flakes of $KC_8$ were transferred to a vial and stored in the freezer. The yield of $KC_8$ was essentially quantitative except for manipulative losses. Due to the high reactivity and pyrophoric nature of $KC_8$, coupled with its lack of spectroscopically diagnostic functionalities, the material was used as prepared in the subsequent synthetic work, without being subjected to characterization techniques. The characteristic golden color of $KC_8$ strongly suggests the presence of the fully reduced reagent and not lower intercalates such as $KC_{24}$ (blue) or $KC_{36}$, $KC_{48}$, or $KC_{60}$ (all black).

### Synthesis of [(salNdipp)₂(Cl)OsN] (1)

Under a $N_2$ atmosphere, a solution of H(salNdipp) (98.7 mg, 0.351 mmol) in 3 ml THF was added over 5 min to a suspension of NaH (14.0 mg, 0.583 mmol, 1.7 eq) in 1 ml THF and stirred vigorously over 90 min, resulting in heavy gas evolution (for several minutes) and a slight darkening of the yellow suspension. Residual NaH was removed by filtration through celite, and the filtrate was transferred to a vial containing (Bu₄N)[Os(N)Cl₄] (103.4 mg, 0.176 mmol) and left to stir overnight, resulting in a slow color change from dark purple to orange. The solvent was removed under reduced pressure, and the residue was redissolved in a mixture of $Et_2O$ and toluene (4.5 ml, 1:2, $Et_2O$:tol), filtered through celite (removing (Bu₄N)Cl and NaCl) and washed with 2 × 2 ml toluene. The solvent was removed under reduced pressure, leaving [(salNdipp)₂(Cl)OsN] (**1**) as an orange solid. Yield of [(salNdipp)₂(Cl)OsN] (**1**): 129.0 mg, 0.161 mmol, 91.7% based on (Bu₄N)[Os(N)Cl₄]. Crystals suitable for X-ray crystallography separated from a hexane solution of **1**, which was concentrated at −35 °C using toluene as a sorbent. **[1]H NMR**, 400 MHz, $C_6D_6$ δ(ppm); 7.90 (s, 1H, imine-CH), 7.86 (s, 1H, imine-CH), 7.27–7.19 (overlapped m, 2H, aryl-CH), 7.19–7.16 (m, 1H, aryl-CH), 7.15–7.09 (overlapped m, 2H, aryl-CH), 7.09–7.00 (overlapped m, 3H, aryl-CH), 6.98 (ddd, $J$ = 8.7, 7.0, 1.8 Hz, 1H, aryl-CH), 6.84 (dd, $J$ = 7.9, 1.8 Hz, 1H, aryl-CH), 6.75 (d, $J$ = 7.2 Hz, 1H, aryl-CH), 6.70 (d, $J$ = 8.5 Hz, 1H, aryl-CH), 6.47 (t, $J$ = 7.4 Hz, 1H, aryl-CH), 6.43 – 6.33 (m, 1H, aryl-CH), 4.63 (hept, $J$ = 6.9 Hz, 1H, $^i$Pr-CH), 4.44 (hept, $J$ = 6.7 Hz, 1H, $^i$Pr-CH), 4.16 (hept, $J$ = 6.8 Hz, 1H, $^i$Pr-CH), 3.15 (hept, $J$ = 6.6 Hz, 1H, $^i$Pr-CH), 1.49 (d, $J$ = 6.7 Hz, 3H, $^i$Pr-CH₃), 1.46 (d, $J$ = 6.7 Hz, 3H, $^i$Pr-CH₃), 1.40 (d, $J$ = 6.7 Hz, 3H, $^i$Pr-CH₃), 1.20 (d, $J$ = 7.2 Hz, 3H, $^i$Pr-CH₃), 1.19 (d, $J$ = 7.2 Hz, 3H, $^i$Pr-CH₃), 1.15 (d, $J$ = 6.7 Hz, 3H, $^i$Pr-CH₃), 1.00 (d, $J$ = 6.8 Hz, 3H, $^i$Pr-CH₃), 0.90 (d, $J$ = 6.6 Hz, 3H, $^i$Pr-CH₃). **[13]C{[1]H} NMR**, 126 MHz, $C_6D_6$ δ(ppm): 172.19, 171.78, 168.67,

166.27, 156.35, 148.13, 144.92, 142.64, 142.64 (*HMBC reveals 2 overlapping peaks at 142.64 ppm*), 141.83, 138.76, 137.58, 137.04, 136.01, 128.65, 128.64 (*HSQC reveals 2 overlapping peaks at 128.65 and 128.64 ppm*), 124.92, 124.29, 124.18, 123.93, 122.11, 121.24, 120.02, 118.48, 117.97, 116.13, 28.21, 27.85, 27.85 (*HSQC reveals 2 overlapping peaks at 27.85 ppm*) 27.57, 26.41, 25.96, 25.64, 24.84, 24.12, 23.92, 23.64, 23.45. **Elemental analysis**, calculated for $C_{38}H_{44}N_3O_2OsCl$: C: 57.02%, H: 5.54%, N: 5.25%; found: C: 56.87%, H: 5.45%, N: 5.21%.

### Synthesis of [(salNdipp)₂(OTf)OsN] (2)

Under a $N_2$ atmosphere, a solution of AgOTf (53.8 mg, 0.209 mmol) in 1 ml toluene was added to a solution of [(salNdipp)₂(Cl)OsN] (**1**, 153 mg, 0.191 mmol) in 2 ml THF under heavy stirring. After 30 minutes, the suspension was filtered through celite (removing AgCl), and the solvents were removed under reduced pressure, leaving [(salNdipp)₂(OTf)OsN] (**2**) as an orange crystalline material. Yield of [(salNdipp)₂(OTf)OsN] · THF (**7**): 180 mg, 0.183 mmol, 95.5% based on **1**. Crystals suitable for X-ray crystallography separated from a hexane solution of **2**, which was concentrated at −35 °C using toluene as a sorbent. **[1]H NMR**, 500 MHz, $C_6D_6$, δ (ppm): 8.00 (s, 2H, imine-CH), 7.43 (dd, $J$ = 7.8, 1.5 Hz, 2H, aryl-CH), 7.30 (t, $J$ = 7.8 Hz, 2H, aryl-CH), 7.10 – 7.03 (overlapped m, 4H, aryl-CH), 7.03 – 6.96 (m, 2H, aryl-CH), 6.77 (dd, $J$ = 7.9, 1.7 Hz, 2H, aryl-CH), 6.38 (ddd, $J$ = 8.0, 6.7, 1.4 Hz, 2H, aryl-CH), 4.00 (hept, $J$ = 6.6 Hz, 2H, $^i$Pr-CH), 2.77 (hept, $J$ = 6.7 Hz, 2H, $^i$Pr-CH), 1.72 (d, $J$ = 6.6 Hz, 6H, $^i$Pr-CH₃), 1.31 (d, $J$ = 6.6 Hz, 6H, $^i$Pr-CH₃), 1.02 (d, $J$ = 6.7 Hz, 6H, $^i$Pr-CH₃), 0.81 (d, $J$ = 6.7 Hz, 6H, $^i$Pr-CH₃). **[13]C{[1]H} NMR**, 101 MHz, $C_6D_6$, δ (ppm): 173.46, 167.40, 146.38, 146.19, 143.30, 139.20, 136.96, 129.34, 124.52, 123.76, 122.26, 119.86, 118.83, 28.79, 27.89, 27.26, 26.02, 22.71, 22.59. **[19]F NMR**, 376 MHz, $C_6D_6$, δ (ppm): −78.17. **UV/Vis**, THF, $λ$ [nm, $ε$ (max/sh, $M^{-1}$ $cm^{-1}$)]: 299 (max, 18700), 391 (max, 4300). **Elemental analysis**, calculated for $C_{39}H_{44}F_3N_3O_5OsS$: C: 51.25%, H: 4.85%, N: 4.60%; found: C: 51.00%, H: 4.81%, N: 4.54%.

### Synthesis of [(salNdipp)₂(DMAP)Os([15]NP)] (3-[15]N)

Complexes **3** and **3-[15]N** were prepared by analogous procedures, here illustrated for the [15]N labeled isotopolog. A solution of H(salNdipp) (116.2 mg, 0.413 mmol) in 3 ml THF was added over 5 min to a suspension of NaH (14.0 mg, 0.583 mmol, 1.4 eq) in 1 ml THF. Vigorous effervescence ensued ($H_2$, several minutes). The suspension was left to stir over 90 min, resulting in a slight darkening of the yellow color. Residual NaH was removed by filtration through celite, and the filtrate was transferred to a vial containing (Bu₄N)[Os([15]N)Cl₄] (121.0 mg, 0.205 mmol) and left to stir overnight, resulting in a slow color change from dark purple to orange. The solvent was removed under reduced pressure, and the residue was redissolved in a mixture of $Et_2O$ and toluene (15 ml, 1:2, $Et_2O$:tol), filtered through celite (removing (Bu₄N)Cl and NaCl) and washed through the filter with 2 × 2 ml toluene. The solvent was removed under reduced pressure, leaving [(salNdipp)₂(Cl)Os[15]N] (**1-[15]N**) as an orange solid. The solid was redissolved in 5 ml THF, and a solution of AgOTf (53.8 mg, 0.209 mmol) in 3 ml toluene was added under vigorous stirring. After 30 minutes, the suspension was filtered through celite (removing AgCl), and the solvents were removed under reduced pressure, leaving [(salNdipp)₂(OTf)Os[15]N] (**2-[15]N**) as an orange crystalline material. The resulting solid and 4-(dimethylamino)pyridine (DMAP, 25.0 mg, 0.205 mmol) were dissolved in 15 ml toluene, and the reaction mixture was left to stir overnight, resulting in an orange suspension. The solvent was removed under reduced pressure. To the solid, Na(OCP) · 2.5 dioxane (68.9 mg, 0.228 mmol, 1.1 eq) in 5 ml THF was added, resulting in evolution of gas (CO, for several minutes) and a color change from light orange to very dark orange. After 30 min, the reaction mixture was filtered through celite, and the solvent was removed under reduced pressure. The solid was then redissolved in a minimal amount of toluene (ca. 30 ml), and $Al_2O_3$ (pH 7, 2.0 g) was added to adsorb NaOTf (this byproduct forms a THF adduct with similar solubility to **3**).

The mixture was stirred for 2 h, the solution was filtered through celite, and the solvent was removed under reduced pressure. The solid residue was redissolved in minimal THF (ca. 5 ml) and diluted with hexane (ca. 50 ml), until the solution started to turn cloudy. The solution was then swirled and left at −35 °C overnight to grow dark orange crystals of [(salNdipp)$_2$(DMAP)Os($^{15}$NP)] (3-$^{15}$N). The mother liquor was removed by decanting, and the very dark orange crystals were washed with cold hexane 3 × 2 ml and dried under reduced pressure. Yield of [(salNdipp)$_2$(DMAP)Os($^{15}$NP)] (3-$^{15}$N): 177.6 mg, 0.193 mmol, 94.1% based on (Bu$_4$N)[Os($^{15}$N)Cl$_4$]. Crystals suitable for X-ray crystallography separated from a hexane solution of 3, which was concentrated at −35 °C using toluene as sorbent. $^1$H NMR, (500 MHz, C$_6$D$_6$) δ: 8.22 (s, 1H, imine-CH), 8.21 (s, 1H, imine-CH), 7.78 (dd, J = 6.9, 1.2 Hz, 1H, DMAP-CH), 7.52 (dd, J = 6.9, 1.1 Hz, 1H, DMAP-CH), 7.42 (dd, J = 7.7, 1.5 Hz, 1H, aryl-CH), 7.22–7.16 (overlapped m, 3H, aryl-CH), 7.16–09 (overlapped m, 2H, aryl-CH), 7.09–7.06 (overlapped m, 3H, aryl-CH), 7.03 (dd, J = 8.5, 1.1 Hz, 1H, aryl-CH), 6.96 (t, J = 7.7 Hz, 1H, aryl-CH), 6.76 (dd, J = 7.7, 1.5 Hz, 1H, aryl-CH), 6.58 (ddd, J = 8.0, 6.9, 1.2 Hz, 1H, aryl-CH), 6.45 (ddd, J = 7.9, 6.7, 1.2 Hz, 1H, aryl-CH), 5.42 (dd, J = 7.0, 3.2 Hz, 1H, DMAP-CH), 5.13 (dd, J = 7.0, 3.2 Hz, 1H, DMAP-CH), 4.96 (hept, J = 6.8 Hz, 1H, $^i$Pr-CH), 4.43 (hept, J = 6.7 Hz, 1H, $^i$Pr-CH), 4.07 (hept, J = 6.7 Hz, 1H, $^i$Pr-CH), 2.00 (s, 6H, DMAP-CH$_3$), 1.94 (hept, J = 6.6 Hz, 1H, $^i$Pr-CH), 1.68 (d, J = 6.6 Hz, 3H, $^i$Pr-CH$_3$), 1.53 (d, J = 6.7 Hz, 3H, $^i$Pr-CH$_3$), 1.38 (d, J = 6.9 Hz, 3H, $^i$Pr-CH$_3$), 1.18 (d, J = 6.7 Hz, 3H, $^i$Pr-CH$_3$), 0.94 (two overlapped d, J = 6.7, 1.3 Hz, 6H, $^i$Pr-CH$_3$), 0.86 (d, J = 6.8 Hz, 3H, $^i$Pr-CH$_3$), 0.59 (d, J = 6.8 Hz, 3H, $^i$Pr-CH$_3$). $^{13}$C{$^1$H} NMR, 126 MHz, C$_6$D$_6$, δ (ppm): 168.85, 166.49, 164.96, 164.84, 152.94, 152.13, 151.92, 150.66, 148.11, 144.85, 143.91, 143.35, 140.58, 135.53, 135.30, 133.88, 133.43, 126.77, 126.57, 124.58, 123.57, 123.41, 123.35, 122.97, 122.71, 121.94, 120.45, 115.22, 114.40, 107.27, 107.02, 38.30, 27.82, 27.74, 26.94, 26.56, 26.30, 26.26, 26.10, 25.95, 25.39, 25.24, 23.16, 22.77. $^{31}$P{$^1$H} NMR, 161.99 MHz, C$_6$D$_6$, δ (ppm): 249.31 (3), 249.02 (3-$^{15}$N, $^1$J$_{PN}$ = 61.8 Hz). $^{15}$N{$^1$H} NMR, 81.11 MHz, C$_6$D$_6$, δ (ppm): 395.97 ($^1$J$_{NP}$ = 62.0 Hz). IR, solid between KBr windows, ν (cm$^{-1}$): 1258/1221 (P≡N) for 3/3-$^{15}$N. UV/Vis, THF, λ [nm, ε (max/sh, M$^{-1}$ cm$^{-1}$)]: 258 (max, 44,900), 318 (max, 25,500), 348 (max, 26,500), 401 (max, 18,000), 572 (sh, 2000). Elemental analysis, calculated for C$_{45}$H$_{54}$N$_5$O$_2$OsP: C: 58.87%, H: 5.93%, N: 7.63%; found: C: 58.86%, H: 5.91%, N: 7.61%.

## Synthesis of [(salNdipp)$_2$(DMAP)Os(NPS$_2$)] (4)

[(salNdipp)$_2$(DMAP)Os(NP)] (3, 25.0 mg, 27.2 μmol) and S$_8$ (1.74 mg, 54.3 μmol of S, 2.0 eq.) (*Note: Sulfur was weighed precisely by making a stock solution of 17.4 mg S$_8$ in 10 ml toluene and taking out a 1 ml aliquot.*) were dissolved in 7.5 ml toluene, resulting in a slow color change from dark orange to dark green. The reaction mixture was cooled to −35 °C overnight, resulting in crystallization of [(salNdipp)$_2$(DMAP)Os(NPS$_2$)] (4) as dark orange crystals. The mother liquor was removed by decanting, and the dark orange crystals were washed with cold toluene (3 × 1 ml) and dried under reduced pressure. Yield of [(salNdipp)$_2$(DMAP)Os(NPS$_2$)] (4), 19.5 mg, 19.9 μmol 72.9% based on 3. Crystals suitable for X-ray crystallography separated from the reaction mixture of [(salNdipp)$_2$(DMAP)Os(NPS$_2$)] (4). $^1$H NMR, 600 MHz, THF-d$_8$, δ (ppm): 7.50 (dd, J = 7.8, 1.8 Hz, 1H, aryl-CH), 7.38 (t, J = 7.7 Hz, 1H, aryl-CH), 7.33 (dd, J = 7.8, 1.5 Hz, 1H, aryl-CH), 7.21 (dd, J = 7.9, 1.8 Hz, 1H, aryl-CH), 7.20 (dd, J = 7.7, 1.6 Hz, 1H, aryl-CH), 7.09 (s, 1H, imine-CH), 7.08 (t, J = 7.8 Hz, 1H, aryl-CH), 7.02 (ddd, J = 8.6, 7.0, 1.8 Hz, 1H, aryl-CH), 6.94 (dd, J = 7.8, 1.5 Hz, 1H, aryl-CH), 6.92 (dd, J = 8.6, 1.1 Hz, 1H, aryl-CH), 6.89 (dd, J = 7.1, 1.3 Hz, 1H, DMAP-CH), 6.86 (ddd, J = 7.9, 6.8, 1.1 Hz, 1H, aryl-CH), 6.73 (dd, J = 7.8, 1.5 Hz, 1H, aryl-CH), 6.65 (dd, J = 7.1, 3.2 Hz, 1H, DMAP-CH), 6.53 (dd, J = 7.1, 1.3 Hz, 1H, DMAP-CH), 6.46 (ddd, J = 8.6, 6.9, 1.8 Hz, 1H, aryl-CH), 6.22 (ddd, J = 7.9, 7.0, 1.1 Hz, 1H, aryl-CH), 6.18 (dd, J = 7.1, 3.2 Hz, 1H, DMAP-CH), 6.06 (dd, J = 8.4, 1.0 Hz, 1H, aryl-CH), 4.93 (s, 1H, imine-CH), 4.49 (hept, J = 6.7 Hz, 1H, $^i$Pr-CH), 3.90 (hept, J = 6.7 Hz, 1H, $^i$Pr-CH), 3.87 (hept, J = 6.7 Hz, 1H, $^i$Pr-CH), 3.14 (hept, J = 6.3 Hz, 1H, $^i$Pr-CH), 3.08 (s, 6H, DMAP-CH$_3$), 1.34 (d, J = 6.8 Hz,

3H, $^i$Pr-CH$_3$), 1.25 (d, J = 6.6 Hz, 3H, $^i$Pr-CH$_3$), 1.24 (d, J = 6.7 Hz, 3H, $^i$Pr-CH$_3$), 1.18 (d, J = 6.7 Hz, 3H, $^i$Pr-CH$_3$), 1.07 (d, J = 6.7 Hz, 3H, $^i$Pr-CH$_3$), 1.01 (d, J = 6.7 Hz, 3H, $^i$Pr-CH$_3$), 0.89 (d, J = 6.7 Hz, 3H, $^i$Pr-CH$_3$), 0.83 (d, J = 6.8 Hz, 3H, $^i$Pr-CH$_3$). $^{13}$C NMR, (151 MHz, THF) δ (ppm): 201.41, 188.72, 184.76, 179.01, 160.01, 156.23, 154.32, 153.95, 150.47, 146.20, 143.65, 142.46, 142.00, 141.98, 140.23, 136.77, 134.92, 128.25, 127.60, 127.60 (HSQC reveals 2 overlapping peaks at 127.60 ppm), 127.25, 124.95, 124.79, 121.65, 120.83, 118.66, 112.77, 111.76, 107.89, 106.00, 105.90, 38.63, 35.93, 29.76, 28.88, 28.02, 27.21, 27.21 (HSQC reveals 2 overlapping peaks at 27.21 ppm), 26.72, 26.23, 23.49, 22.97, 22.88, 22.48. $^{31}$P{$^1$H} NMR, 243 MHz, THF-d$_8$, δ (ppm): 233.13 (4), 232.64 (4-$^{15}$N, d, $^1$J$_{PN}$ = 50.6 Hz). $^1$H-$^{15}$N HMBC NMR, 61 MHz, THF-d$_8$, δ (ppm): 954.53 (d, $^1$J$_{NP}$ = 50.5 Hz). UV/Vis, THF, λ [nm, ε (max/sh, M$^{-1}$ cm$^{-1}$)]: 308 (max, 29000), 382 (max, 28000), 440 (sh, 12000), 512 (max, 5900), 580 (max, 4800). Elemental analysis, calculated for C$_{45}$H$_{54}$N$_5$O$_2$OsPS$_2$: C: 55.02%, H: 5.54%, N: 7.13%; found: C: 54.74%, H: 5.59%, N: 7.05%.

## Synthesis of [(salNdipp)$_2$(DMAP)Os(NPCl)] (5)

[(salNdipp)$_2$(DMAP)Os(NP)] (3, 100.0 mg, 0.109 mmol) and Ph$_3$CCl (40.5 mg, 0.145 mmol, 1.3 eq.) were dissolved in 5 ml THF, and the reaction mixture was left for 3 hours. The solvent was removed under reduced pressure, and the dark orange residue was redissolved in a minimum amount of THF (3 ml), diluted with hexane (15 ml), cooled to −35 °C, and left to crystallize overnight. The mother liquor was removed by decanting, and the dark orange crystals of [(salNdipp)$_2$(DMAP)Os(NPCl)] (5) were washed with cold hexane (3 × 2 ml), and dried under reduced pressure. Yield of [(salNdipp)$_2$(DMAP)Os(NPCl)] (5): 100.7 mg, 0.106 mmol, 97.0% based on 3. Crystals suitable for X-ray crystallography separated from a THF solution of [(salNdipp)$_2$(DMAP)Os(NPCl)] (5), with hexane/toluene as sorbent at −35 °C. $^1$H NMR (400 MHz, C$_6$D$_6$) δ 23.31 (FWHM = 390 Hz), 19.95 (FWHM = 190 Hz), 18.69 (FWHM = 200 Hz), 18.11 (FWHM = 130 Hz), 16.03 (FWHM = 120 Hz), 10.23 (FWHM = 120 Hz), 10.05 (FWHM = 60 Hz), 9.20 (FWHM = 50 Hz), 8.31 (FWHM = 40 Hz), 7.38 (FWHM = 20 Hz), 7.35 (FWHM = 5 Hz), 6.95 (FWHM = 10 Hz), 5.33 (FWHM = 30 Hz), 3.83 (FWHM = 140 Hz), 3.62 (FWHM = 30 Hz), 3.13 (FWHM = 90 Hz), 1.93 (FWHM = 60 Hz), 0.56 (FWHM = 70 Hz), −0.03 (FWHM = 60 Hz), −0.60 (FWHM = 90 Hz), −2.86 (FWHM = 50 Hz), −3.48 (FWHM = 60 Hz), −7.48 (FWHM = 300 Hz), −10.64 (FWHM = 350 Hz), −22.24 (FWHM = 1100 Hz). UV/Vis, THF, λ [nm, ε (max/sh, M$^{-1}$ cm$^{-1}$)]: 256 (max, 29,000), 337 (max, 17,000), 395 (max, 13,000). Magnetic moment, μ$_{eff}$ (Evans' method, THF-d$_8$, 298 K): 1.83 μ$_B$. Elemental analysis, calculated for C$_{45}$H$_{54}$ClN$_5$O$_2$OsP: C: 56.68%, H: 5.71%, N: 7.34%; found: C: 56.59%, H: 5.74%, N: 7.31%.

## Synthesis of [(salNdipp)$_2$(DMAP)Os($\eta^1$-N$_4$P)] (6)

[(salNdipp)$_2$(DMAP)Os(NPCl)] (5, 25.0 mg, 26.2 μmol) and Me$_3$SiN$_3$ (3.44 mg, 29.8 μmol, 1.15 eq) were dissolved in 1.5 ml THF, and the reaction mixture was left for 1 h. The solution was diluted with hexane (ca. 10 ml) to precipitate out dark orange crystals of [(salNdipp)$_2$(DMAP)Os($\eta^1$-N$_4$P)] (6), the mixture was cooled to −35 °C for 30 min and washed with 3 × 1 ml hexane. Yield of [(salNdipp)$_2$(DMAP)Os($\eta^1$-N$_4$P)] (6), 13.9 mg, 14.5 μmol, 55.2% based on 5. Crystals suitable for X-ray crystallography separated from a Et$_2$O solution of [(salNdipp)$_2$(DMAP)Os($\eta^1$-N$_4$P)] (6), with toluene as sorbent at −35 °C. $^1$H NMR (400 MHz, C$_6$D$_6$) δ 18.87 (FWHM = 80 Hz), 17.16 (FWHM = 100 Hz), 17.09 (FWHM = 60 Hz), 14.62 (FWHM = 100 Hz), 13.60 (FWHM = 40 Hz), 12.36 (FWHM = 40 Hz), 10.99 (FWHM = 40 Hz), 9.08 (FWHM = 70 Hz), 8.29 (FWHM = 20 Hz), 7.65 (FWHM = 20 Hz), 7.63 (FWHM = 20 Hz), 6.63 (FWHM = 20 Hz), 5.65 (FWHM = 60 Hz), 5.35 (FWHM = 70 Hz), 3.75 (FWHM = 30 Hz), 3.00 (FWHM = 40 Hz), 1.76 (FWHM = 60 Hz), 0.97 (FWHM = 20 Hz), −1.04 (FWHM = 110 Hz), −1.81 (FWHM = 50 Hz), −5.60 (FWHM = 40 Hz), −5.76 (FWHM = 30 Hz), −6.38 (FWHM = 200 Hz), −14.01 (FWHM = 550 Hz), −19.64 (FWHM = 270 Hz),

−23.41 (FWHM = 500 Hz), −30.42 (FWHM = 1500 Hz). **UV/vis**, THF, $\lambda$ [nm, $\varepsilon$ (max/sh, M$^{-1}$ cm$^{-1}$)]: 320 (max, 19,000), 390 (max, 12,000), 592 (max, 1500). **Magnetic moment**, $\mu_{eff}$ (Evans' method, THF-$d_8$, 298 K): 1.85 $\mu_B$. **Elemental analysis**, calculated for C$_{45}$H$_{54}$N$_8$O$_2$OsP: C: 56.29%, H: 5.67, N: 11.67; found: C: 56.19%, H: 5.65%, N: 11.64%.

*Note: other azide sources also convert **5** to **6**, but due to the low stability of the product, these methods do not allow a pure product to be isolated. [1] When using (Bu$_4$N)(N$_3$) in C$_6$D$_6$, **6** forms cleanly within 5 minutes, but the (Bu$_4$N)Cl byproduct has a similar solubility to **6**; attempts at removing (Bu$_4$N)Cl with alumina resulted in full decomposition of **6**. [2] When using NaN$_3$ in THF or dioxane, the formation of **6** is so slow that its thermal decomposition to **3** and N$_2$ complex **7** prevents isolation of a pure product. [3] When using NaN$_3$ with LiCl as phase-transfer catalyst in THF, the conversion rate is variable, and lithium ion remains in the sample, as verified from $^7$Li NMR.*

## Data availability

All relevant data generated in this study, including full experimental procedures, crystal structures, magnetic data, $^1$H, $^{13}$C, $^{15}$N, $^{19}$F, $^{31}$P, COSY, HSQC, and HMBC 2D NMR data, IR, UV-vis, and XANES spectral data, X-ray crystallographic data, as well as DFT calculations, are included in this Article, its Supplementary Information, and Source Data file. The X-ray crystallographic coordinates for structures reported in this study have been deposited at the Cambridge Crystallographic Data Center (CCDC), under deposition numbers CCDC 2388544 (**1**), 2388542 (**2**), 2388539 (**3**), 2388543 (**4**), 2388541 (**5**), 2388540 (**6**), 2388538 (**7**). These data can be obtained free of charge from The Cambridge Crystallographic Data Center via www.ccdc.cam. ac.uk/data_request/cif. All data are available from the corresponding author upon request. Source data are provided with this paper.

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

## Acknowledgements

The authors thank The Carlsberg Foundation (reintegration fellowship, CF21-0438, A.R.), The Swedish Research Council (2022-03154, A.R.), The Royal Physiographical Society of Lund (A.R.), Stiftelsen Lars Hiertas Minne (FO2022-0033, A.R.), The Crafoord Foundation (20230776, A.R.), The Carlsberg Foundation (research infrastructure grant, CF17-0637, K.S.P.), and the Danish Agency for Science, Technology, and Innovation for funding the instrument center "Danscatt" (K.S.P.). The X-ray spectroscopy experiments were performed at the ID12 beamline at the European Synchrotron Radiation Facility (Grenoble, France). The authors thank UTEP HPC JAKAR Cluster for the computational resources provided free of charge (C.S.-P. and B.P.). The views expressed are purely those of the authors and may not in any circumstances be regarded as stating an official position of the ERCEA and the European Commission.

## Author contributions

A. Reinholdt conceived and planned the research project. S.E. and A. Reinholdt carried out the synthetic work. S.E. carried out and analyzed the UV-vis spectroscopic, IR spectroscopic, and X-ray crystallographic studies. S.E. and Z.T. carried out and analyzed the NMR spectroscopic studies. C.S.-P. and B.P. carried out and analyzed the computational DFT studies. N.J.Y., F.W., A. Rogalev, and K.S.P. carried out and analyzed the X-ray spectroscopic studies. K.S.P. carried out and analyzed the studies of the magnetic properties. All authors contributed to writing the paper.

## Funding

## Competing interests

The authors declare no competing interests.
