## [Transparent Peer Review file · Nature Communications]

Unleashing Phosphorus Mononitride

Corresponding Author: Dr Anders Reinholdt

Version 0:

Reviewer comments:

Reviewer #1

(Remarks to the Author)

This work by Reinholdt et al. reported a new method for the construction of metal-bound phosphorus mononitride via P-atom transfer reaction of Na(OCP) with an electrophilic osmium nitride complex. Quite interestingly, the oxidation reactions of the metal-bound PN have been explored, and a novel aromatic interpnictide [PN₄]⁻ was synthesized from the and 3+2 cycloaddition of the [NPCl]⁻ with an azide. All the new compounds were fully characterized by combining spectroscopic methods and X-ray crystallography, and the bonding properties of the P-N multiple bonding properties were analyzed. The diatomic molecule PN is very reactive, and its chemistry has remained barely known. I was very much impressed by the new PN chemistry in the present study, and the multiple bonding properties displayed in the new PN-containing compound are also of fundamental importance in main-group chemistry. The experiments were well designed, and all the compounds were unambiguously characterized. I recommend publication of this exciting work in Nature Communications with two minor suggestions:

(1) The mechanism for the thermal decomposition of 6 is very interesting. The formation of 7 from 6 requires the concomitant generation of PNN, which may further react to produce N₂ and P₃N₅ in the solution. This means that the selective cleavage of the [PN₄] ring (→ PN + N₃; → N₂ + PNN; → N₂ + NPN) may provide unique synthetic access to the highly elusive PNN moiety, which is a phosphorus analog of N₃. To understand the thermal stability of this five-membered ring, theoretical calculations on the barriers for the three decomposition pathways would be helpful. Experimentally, some chemical trapping reactions of the PN₃ moiety could be performed. Alternatively, detection of the gaseous decomposition products with mass spectrometry could provide more evidence for the fragments. Additionally, a recent work about the low-temperature synthesis of HPNN and the PN₂ radical was reported by the photoreaction of HPCO with N₂ (J. Am. Chem. Soc. 2022, 144, 21853–21857).

(2) Page 4, it was mentioned that the P-Cl bond in compound 5 is one of the longest structurally characterized P-Cl bonds (top 1%). Please add a representative reference for the comparison.

Reviewer #2

(Remarks to the Author)

S. Edin and collaborators present a piece of work on inorganic synthesis of PN multiple bonds. In my opinion the article is of good quality, innovative and of a relevance that would attract the readers of Nature Communications. However, it is my opinion that the material fails to address the one of the most important points. The implications of such synthetic breakthrough together with the limitations and wider applicability of the synthetic method developed. I suggest that this was to be fully addressed, then it could be considered for publication

1. In the introduction, authors state that neutral species B, C and D from chart 1, are more closely related to a P-N description rather than to that of a triple bond. Authors should consider expanding this point and clearly stating the reasons for such comment. This is key for the readers to have an understanding of the characteristics that the authors are "looking for" when trying to generate the PN triple bonds

2. In the same section, authors mention the synthesis of iron complex F. As a half-sandwich/ piano-stool complex, authors should probably include information on the stability of this compound and whether the NP monodentate ligand is susceptible to hydrolysis.

3. Authors summarise their work in scheme 1, at the start of the results and discussion section. I would suggest that authors include here their observations on the stability of each of the complexes (1 to 6) and how critical their choices of temperature and atmosphere were for the synthetic procedures

4. Authors should include reasons behind the similarities/differences of the chemical shifts observed when compared to compounds in Chart 1. Otherwise, there is no much point in just stating those. When discussing the absorption bands in UV-Vis, authors should also include the conditions in which that spectrum was obtained.
5. Figure 2 needs more details in the caption. This should be fully understandable from a stand alone perspective, without having the text to have the details
6. Formation of compounds 4 and 5 should also explicitly discuss the implications/advantages/disadvantages of their experimental details
7. Authors mention the metastable nature of 6. This point definitely needs to be further argued in more detail
8. As above, caption for Figure 4 needs enough detail for a stand alone piece
9. The conclusions of this study fail to evaluate the implications of this synthetic breakthrough. This needs to also include the limitations and advantages based on their experimental choices, as well as, how much of this can really be translated beyond these 7 compounds.
10. Elemental analysis for compounds 4 and 6 in particular are quite far off from the expected values. Is there an explanation for this?
11. In the reference section, pay special attention to entries number. They have formatting issues that need to be addressed

Reviewer #3

(Remarks to the Author)

[From editor: please see attached]

Version 1:

Reviewer comments:

Reviewer #1

(Remarks to the Author)

The authors addressed my concerns, and the manuscript has been improved after making revisions according to the comments. Now, I recommend its publication on Nature Communications in its current form.

Reviewer #2

(Remarks to the Author)

IT is my opinion that the authors have now addressed the comments/suggestions/questions from all three reviewers. New data has been included to support and expand several of the points highlighted. Therefore, I am satisfied that the requirements have been fullfield that that the publication can now take place.

Reviewer #3

(Remarks to the Author)

The revision of the manuscript was carried out very carefully and in detail. My critical comments were also addressed in great detail. However, as far as my criticism of novelty is concerned, I cannot really see much that is new. Of course, the novelty value of a publication is not a physical quantity and is always subject to a subjective factor, but I stand by my criticism:

- (i) PN / HPN known (by the way, ACIE e202414456 should be cited)
- (ii) OCP known as P-transfer reagent
- (iii) MNP known (M = metal, and yes M = Os is new)
- (iv) 2+3 cycloaddition of NP + N3 fragment to give a N4P heterocycle known
- (v) N4P heterocycle motif known

The authors rightly point out that an Os-N bond (in OsN4P) is certainly to be considered somewhat differently than a C-N bond (in C-N4P) and further assume that it is correct to equate this with N4P(-) and compare it with P5(-) and N5(-). Here I would like to disagree since both N5(-) and P5(-) have actually been synthesized almost naked, namely in the presence of a weakly coordinating cation! This is the reason, for example, why P5(-) has only now (2025!) been isolated as a "naked" anion for the first time by Müller in the solid state (see ACIE <https://doi.org/10.1002/anie.202505853>, which maybe should be cited and discussed). The authors then point out in their reply that it is a dative Os-N bond, if I understand them correctly. Again, I would disagree, as they show no experiment in which N4P(-) is actually released (e.g. in solution) or separated from osmium (unlike N5(-) and P5(-)).

And even the IUPAC emphasizes: "The synonym 'dative bond' is obsolete. (The origin of the bonding electrons has by itself no bearing on the character of the bond formed. Thus, the formation of methyl chloride from a methyl cation and a chloride ion involves coordination; the resultant bond obviously differs in no way from the C-Cl bond in methyl chloride formed by any other path, e.g. by colligation of a methyl radical and a chlorine atom.)" IUPAC Compendium of Chemical Terminology, 5th ed. International Union of Pure and Applied Chemistry; 2025. Online version 5.0.0, 2025. <https://doi.org/10.1351/goldbook.C01329>.

So, for me, the C-N bond in R-N4P is a polar covalent bond, as is the Os-N bond in Ligand-Os-N4P, unless the authors

show experimentally that they observe (almost) naked N5P(-) in solution or in the solid state. Neither has happened. The claim to have generated somewhat N4P(-) (in the manuscript: "definitively be viewed as [PN4]- bound to OsIII") is wrong to my mind.

Furthermore, from my point of view, the X-ray structure (OsN4P) with a Rint value of 0.2489 (see IURC recommendation below) is unacceptable. This would not be possible in my working group. The fact that the Rint value is much too large and almost twice as large as the wr2 value indicates problems in the solution and should actually lead to warnings in the checkcif, which I unfortunately do not have now. I would not allow publication with such bad R-values for my people. (According to IUCr: In the IUCr checkCIF procedure, the Rint value should usually be less than 0.12. Higher values may indicate issues with the data or the crystal.)

In addition to the above-mentioned points, which led to my conclusion that I am not in favor of an acceptance of this manuscript in Nature Comm, there are also things that I still see differently, but I am of the opinion that certain exp. and theor. data can and may well be discussed differently. For example, the oxidation states, where I still think that it is not reasonable to have a difference of 4 electrons Os(II) to Os(VI) in a resonance scheme. Or the comparison with PN in the universe. Even if other colleagues have done so, this is not an argument for me, nor is it the comparison with N2 or CO, which are known to thermally dissociate easily, which was not shown for PN. I'm just old school and love down-to-earth introductions 😊.

To be fair, since two out of three reviewers voted very enthusiastically for this manuscript and I myself as an author have always been very happy when the editor then decided in my favor, I would like to encourage the editor to decide in favor of acceptance here as well, even if I see it differently (see my points (i)-(v)). Why? It is still beautiful chemistry, and the excellent molecular chemistry described here was carried out competently, and I greatly appreciate how difficult the experiments were to be conducted!

Reviewer #1 (Remarks to the Author)

This work by Reinholdt et al. reported a new method for the construction of metal-bound phosphorus mononitride via P-atom transfer reaction of Na(OCP) with an electrophilic osmium nitride complex. Quite interestingly, the oxidation reactions of the metal-bound PN have been explored, and a novel aromatic interpnictide $[\text{PN}_4]^-$ was synthesized from the and 3+2 cycloaddition of the $[\text{NPCI}]^-$ with an azide. All the new compounds were fully characterized by combining spectroscopic methods and X-ray crystallography, and the bonding properties of the P-N multiple bonding properties were analyzed. The diatomic molecule PN is very reactive, and its chemistry has remained barely known. I was very much impressed by the new PN chemistry in the present study, and the multiple bonding properties displayed in the new PN-containing compound are also of fundamental importance in main-group chemistry. The experiments were well designed, and all the compounds were unambiguously characterized. I recommend publication of this exciting work in *Nature Communications* with two minor suggestions:

We are delighted that Reviewer #1 finds our reactivity studies interesting and points out the novelty of these transformations in the recommendation to publish our study in *Nature Communications*.

(1) The mechanism for the thermal decomposition of **6** is very interesting. The formation of **7** from **6** requires the concomitant generation of PNN, which may further react to produce N_2 and P_3N_5 in the solution. This means that the selective cleavage of the $[\text{PN}_4]$ ring ($\rightarrow \text{PN} + \text{N}_3$; $\rightarrow \text{N}_2 + \text{PNN}$; $\rightarrow \text{N}_2 + \text{NPN}$) may provide unique synthetic access to the highly elusive PNN moiety, which is a phosphorus analog of N_3 . To understand the thermal stability of this five-membered ring, theoretical calculations on the barriers for the three decomposition pathways would be helpful. Experimentally, some chemical trapping reactions of the PN_3 moiety could be performed. Alternatively, detection of the gaseous decomposition products with mass spectrometry could provide more evidence for the fragments. Additionally, a recent work about the low-temperature synthesis of HPNN and the PN_2 radical was reported by the photoreaction of HPCO with N_2 (J. Am. Chem. Soc. 2022, 144, 21853–21857).

We thank Reviewer #1 for sharing these insights on possible modes of rupture of the $[\text{PN}_4]^-$ ring, which we have addressed in more detail in the revised manuscript. We would like to preface this section by noting that the previous report on the observation of the extremely reactive PNN^* radical required very low temperature (e.g. Ar matrix, 10 K), and as such, we would not be surprised if the radicals, PNN^* , NPN^* , or N_3^* , might elude observation at room temperature.

[1]

Our computational studies revealed that decomposition of **6** to form **7** and PNN^* ($\Delta G^\circ = -2.34$ kcal/mol) is the only thermodynamically viable process and, thus, the most plausible scenario among the considered mechanisms. Loss of NPN^* or N_3^* was found to be endothermic processes (+57.37 and +35.87 kcal/mol, respectively). Given that these new data, as predicted by Reviewer #1, suggest the intermediacy of PNN^* , we have cited Zeng and co-workers in the revised manuscript.

Pathway	$\Delta G^\circ(\text{sol}) - \text{TPSSh}$ (kcal/mol)
$[(\text{salNdipp})_2(\text{DMAP})\text{Os}(\eta^1\text{-N}_4\text{P})] \rightarrow [(\text{salNdipp})_2(\text{DMAP})\text{Os}(\text{NP})] + [\text{N}_3]^*$	+35.87
$[(\text{salNdipp})_2(\text{DMAP})\text{Os}(\eta^1\text{-N}_4\text{P})] \rightarrow [(\text{salNdipp})_2(\text{DMAP})\text{Os}(\text{N}_2)] + [\text{PNN}]^*$	-2.34
$[(\text{salNdipp})_2(\text{DMAP})\text{Os}(\eta^1\text{-N}_4\text{P})] \rightarrow [(\text{salNdipp})_2(\text{DMAP})\text{Os}(\text{N}_2)] + [\text{NPN}]^*$	+57.37

[2]

We attempted to detect the PNN[•] radical by analyzing THF/toluene solutions of **6** and radical traps (*N-tert-butyl- α -phenylnitrone*, 5,5-dimethyl-1-pyrroline *N-oxide*) by EPR. Initially, ¹H NMR spectroscopy showed no reaction between **6** and either trap. However, we could not identify any trapped radical products by EPR spectroscopy.

[3]

We also conducted a thermogravimetry-mass spectrometry analysis of **6** stored in a stream of argon. The sample lost about 6% mass from 100–150 °C. The carrier gas was analyzed in a mass spectrometer, but we could not establish any *m/z* fragments corresponding to N₃[•] or PNN[•]/NPN[•].

[4]

Finally, we extended our IR analysis of products formed when **6**/**6**-¹⁵N decompose in C₆D₆ (**6**-¹⁵N prepared from ¹⁵N-labelled osmium nitride). After having removed the solvent, we observed resonances at 1258/1221 cm⁻¹ (**3**/**3**-¹⁵N) and at 2021/1988 cm⁻¹ (**7**/**7**-¹⁵N). The isotopic shift for **7**/**7**-¹⁵N is in close agreement with the value expected for a ¹⁴N \equiv ¹⁴N/¹⁵N \equiv ¹⁴N harmonic oscillator (both 1.017). Importantly, the decomposition mixtures obtained from **6**-¹⁵N show no resonances attributable to unlabeled P \equiv ¹⁴N or ¹⁴N₂ ligands. This suggests that the ¹⁵N-label remains coordinated to osmium throughout the decomposition process; it is unlikely that the [PN₄]⁻ heterocycle slides in such a way that it attains a higher hapticity, and it can be ruled out that the N₂ ligand in **7** derives from the atmosphere (all experiments were carried out under N₂). In essence, isotopic labeling indicates that the Os–N bond does not break during the decomposition process.

(2) Page 4, it was mentioned that the P–Cl bond in compound **5** is one of the longest structurally characterized P–Cl bonds (top 1%). Please add a representative reference for the comparison.

We have referenced the CSD, from which we have extracted statistics on P–Cl bond distances. *Acta Crystallogr., Sect. B* **72**, 171-179 (2016). Moreover, we have referenced the very long P–Cl bond found in the [PCl₄]⁻ ion [2.850(4) Å], which defines the high end of the P–Cl bond distance range. *Z. Anorg. Allg. Chem.* **488**, 7-26 (1982).

Reviewer #2 (Remarks to the Author)

S. Edin and collaborators present a piece of work on inorganic synthesis of PN multiple bonds. In my opinion the article is of good quality, innovative and of a relevance that would attract the readers of *Nature Communications*. However, it is my opinion that the material fails to address the one of the most important points. The implications of such synthetic breakthrough together with the limitations and wider applicability of the synthetic method developed. I suggest that this was to be fully addressed, then it could be considered for publication

We thank Reviewer #2 for such constructive feedback and for pointing out the need to emphasize the implications and wider applicability of our synthetic methodology. The original version of the manuscript misses this aspect to some extent – we have made an extra effort in the revised version to describe the implications and general applicability of the synthetic method that we developed to obtain these rare species.

1. In the introduction, authors state that neutral species **B**, **C** and **D** from chart 1, are more closely related to a P–N description rather than to that of a triple bond. Authors should consider expanding this point and clearly stating the reasons for such comment. This is key for the readers to have an understanding of the characteristics that the authors are “looking for” when trying to generate the PN triple bonds

This is obviously a central feature that deserves to be highlighted. We have explained in more detail how the P–N bonding in **B** has competing carbene C=P and N=C double bond character, while the three-coordinate nature of the P and N atoms in **C** and **D** results in these species having predominantly P–N single bond character.

2. In the same section, authors mention the synthesis of iron complex **F**. As a half-sandwich/ piano-stool complex, authors should probably include information on the stability of this compound and whether the NP monodentate ligand is susceptible to hydrolysis.

In the case of compound **E**, light-induced linkage isomerism was observed by *in-crystallo* X-ray diffraction studies (Mo–NP to Mo–PN), but a similar situation was not observed for **F**, for which only the N-bound isomer was experimentally observed. In the revised manuscript, we have mentioned how calculations show the N-bonded linkage isomer of **F** to be energetically preferred over the P-bound and side-on bound isomers by 14.3 and 16.8 kcal mol⁻¹. It is fascinating that P≡N shows a preference for binding through N, even when transferred in solution from a P≡N-releasing precursor and onto the iron center in **F**.

As for the susceptibility of monodentate P≡N toward hydrolysis, we are not aware of previous studies that have addressed this fundamental type of reactivity. However, we have now confirmed that our complex **3** decomposes to an intractable mixture of compounds when a droplet of water is added to a solution of **3** in C₆D₆ (under N₂ atmosphere), which demonstrates that the Os-bound P≡N ligand is susceptible to hydrolysis.

3. Authors summarise their work in scheme 1, at the start of the results and discussion section. I would suggest that authors include here their observations on the stability of each of the complexes (**1** to **6**) and how critical their choices of temperature and atmosphere were for the synthetic procedures.

Due to this recommendation of Reviewer #2, with which we fully agree to be relevant for future research, we incorporated the key results of standard stability studies on **1-6** in the revised manuscript (added to Scheme 1).

1 is stable in air. It is thermally stable when dissolved in C₆D₆ under N₂ (80 °C, 18 hours).

2 decomposes in air. It is thermally stable when dissolved in C₆D₆ under N₂ (80 °C, 18 hours).

3 decomposes in air; hydrolysis sensitive. It is thermally stable in C₆D₆ under N₂ (120 °C, 24 hours).

4 decomposes in air. It is thermally stable when dissolved in C₆D₆ under N₂ (80 °C, 18 hours).

5 decomposes in air. It is thermally stable when dissolved in C₆D₆ under N₂ (80 °C, 18 hours).

6 decomposes in air. In the solid state, **6** is unstable at –35 °C and slowly decomposes over the course of weeks. The compound fully decomposes when dissolved in a sealed tube (J Young) in C₆D₆ under N₂ (50 °C, 18 hours; or at 25 °C over a few days).

4. Authors should include reasons behind the similarities/differences of the chemical shifts observed when compared to compounds in Chart 1. Otherwise, there is no much point in just stating those. When discussing the absorption bands in UV-Vis, authors should also include the conditions in which that spectrum was obtained.

We thank Reviewer #2 for this constructive feedback, which has enhanced the clarity of the spectroscopic paragraph. We have revised this section so it explains more succinctly how the spectroscopic differences unveil different degrees of π -backdonation from the metal centers (Fe^{II}, Mo^{II}, Os^{II}) to the P≡N ligands. We have also added information about solvent (THF) and concentration for the UV-vis data in the manuscript text and the caption of Figure 1.

5. Figure 2 needs more details in the caption. This should be fully understandable from a stand alone perspective, without having the text to have the details

We have expanded the caption such that the key computational details (DFT, TPSSh-D3/def2-TZVP, and isovalue) can be directly accessed, and we have also described the main conclusions about the cumulenic nature of **3** that can be deduced from the MOs.

6. Formation of compounds **4** and **5** should also explicitly discuss the implications/advantages/disadvantages of their experimental details

To clarify this, we have added details about the particular choice of reaction conditions for the sulfur transfer reaction (S_8 and **3** being soluble in toluene, **4** crystallizing directly from the reaction mixture, and the reversible interconversion between **3** and **4**). Likewise, we have added information on the chlorine transfer reaction (Ph_3CCl being an easily handled and mild oxidant, forming a hexane-soluble by-product that is easy to remove, and, most importantly, how this particular choice of oxidant avoids any over-oxidation of the $[Os=N=P]$ system). The utility of **5** as a starting material is illustrated in the paragraph devoted to **6**, and we see this as the most suitable place to discuss the synthetic implications of **5**.

7. Authors mention the metastable nature of **6**. This point definitely needs to be further argued in more detail

This is a central question, which was also brought up by Reviewer #1. The two reviewers' suggestions have prompted us to conduct additional spectroscopic experiments and computational studies, which have cast more light on the metastable nature of **6**. We reproduce our central findings here for clarity:

[1]

In the original manuscript, we observed that **6** decomposes in C_6D_6 solution to form $P\equiv N$ complex **3** and N_2 complex **7** in a 1:1 proportion. We have now gone one step further, analyzed the non-volatile decomposition products by IR spectroscopy, and compared the results to the outcome for when an isotopically labelled heterocycle, **6**- ^{15}N , decomposes. The IR spectra clearly reveal **3/3**- ^{15}N from their $P\equiv N$ stretching modes at $1258/1221\text{ cm}^{-1}$. More interestingly, the formation of N_2 complexes **7/7**- ^{15}N could be ascertained from stretching modes at $2021/1988\text{ cm}^{-1}$; the isotopic shift between these isotopologues agrees closely with the calculated value for a $^{14}N\equiv^{14}N/^{15}N\equiv^{14}N$ harmonic oscillator (1.017). This simple experiment demonstrates the retention of the ^{15}N label in the decomposition products. We therefore conclude that the $Os-N_4P$ linkage does not break in the decomposition process. We also conclude that the N_2 ligand in **7** derives from the $[PN_4]^-$ ring, and not from the atmosphere, despite the experiments being conducted under N_2 . Furthermore, it is unlikely that the $[PN_4]^-$ ring slides to attain higher hapticity during the decomposition pathway.

[2]

We also conducted computational studies to evaluate the most likely decomposition pathways for **6**. From considering unimolecular pathways, we could see that a retro [3+2] cycloaddition forming **3** and a radical such as PNN^* is the only energetically favorable transformation ($\Delta G^0 = -2.34\text{ kcal/mol}$). Loss of NPN^* or N_3^* was found to be endothermic processes ($+57.37$ and $+35.87\text{ kcal/mol}$, respectively).

Pathway	$\Delta G^{\circ}(\text{sol}) - \text{TPSSh}$ (kcal/mol)
$[(\text{salNdipp})_2(\text{DMAP})\text{Os}(\eta^1\text{-N}_4\text{P})] \rightarrow [(\text{salNdipp})_2(\text{DMAP})\text{Os}(\text{NP})] + [\text{N}_3]^{\bullet}$	+35.87
$[(\text{salNdipp})_2(\text{DMAP})\text{Os}(\eta^1\text{-N}_4\text{P})] \rightarrow [(\text{salNdipp})_2(\text{DMAP})\text{Os}(\text{N}_2)] + [\text{PNN}]^{\bullet}$	-2.34
$[(\text{salNdipp})_2(\text{DMAP})\text{Os}(\eta^1\text{-N}_4\text{P})] \rightarrow [(\text{salNdipp})_2(\text{DMAP})\text{Os}(\text{N}_2)] + [\text{NPN}]^{\bullet}$	+57.37

8. As above, caption for Figure 4 needs enough detail for a stand alone piece

We have expanded the caption such that more experimental details on the XANES study can be directly accessed. We have also explained how XANES enables us to draw central conclusions about the d-electron population of the osmium complexes, therefore providing a spectroscopic probe of the oxidation state of the metal centers.

9. The conclusions of this study fail to evaluate the implications of this synthetic breakthrough. This needs to also include the limitations and advantages based on their experimental choices, as well as, how much of this can really be translated beyond these 7 compounds.

Thank you very much! Understanding the importance and addressing this remark of Reviewer #2 significantly improved the overall impact and conclusions of our study. In short, we mention:

[1] That the P-atom transfer from Na(OCP) to an electrophilic osmium nitride can likely be generalized to many other nitride complexes, giving access to P \equiv N ligands bound to a variety of transition metals. We expect that electrophilic character of the nitrides is going to be key to the success of the P-atom transfer strategy introduced in our study. Hence, late transition metals are likely platforms for further studies of P \equiv N chemistry.

[2] We expect that **4** will show group transfer reactivity in which its two sulfur atoms can be exchanged for isovalent moieties, therefore serving as a versatile chemical precursor to PN multiply-bonded main-group motifs.

[3] In a similar vein, we expect that halide exchange on **5** will prove a facile entry not only to exotic heterocycles (as shown in our study), but also to a broad range of unusual multiply-bonded [NPX]⁻ motifs, and perhaps even a radical cation – we are pursuing all of these possibilities and hope to communicate them in due course.

10. Elemental analysis for compounds **4** and **6** in particular are quite far off from the expected values. Is there an explanation for this?

We thank Reviewer #2 for rigorously scrutinizing both the manuscript and the Supplementary Information.

We unfortunately fail to follow Reviewer #2 here in relation to the analysis of complex **4**. The elemental analysis of **4** shows that all measured values for C, H, and N are within 0.3% of the calculated value. This shows that the bulk elemental composition is in agreement with the chemical structure of **4**. *Nature Communications'* guidelines for the characterization of chemical and biomolecular materials state that elemental analyses should be within $\pm 0.4\%$ of the calculated value.

As for the elemental analysis of **6**, our first elemental analysis was indeed sub-optimal (which we explicitly stated in the Supplementary Information and Methods section). For this particular compound, it is important to consider that **6** is thermally unstable. The onset of decomposition of crystalline samples of **6** can be seen after a few days, even when the crystalline material is stored in the freezer of our glovebox ($-35\text{ }^{\circ}\text{C}$). Therefore, **6** has surely undergone thermal decomposition on the way to the analysis laboratory (Kolbe, Germany). However,

during the revisions, we repeated the synthesis of **6** in triplicate and submitted these samples for elemental analysis. This resulted in acceptable analyses: calc. for $C_{45}H_{54}N_8O_2OsP$: C: 56.29%, H: 5.67, N: 11.67; found: C: 56.19%, H: 5.65%, N: 11.64%.

11. In the reference section, pay special attention to entries number. They have formatting issues that need to be addressed.

We have invested significant effort to identify any formatting issues in the References. We realized that the reference numbers were not followed by a full stop, and we have now added this punctuation. Otherwise, we have written the citations in keeping with the house style of *Nature Communications*:

Author list, Title. *Journal* **Volume**, pages (year)

In compliance with this style, all authors are included in the reference lists unless there are more than five, in which case only the first author is given, followed by 'et al.'.

Reviewer #3 (Remarks to the Author)

The research groups of Reinholdt and Pederson describe a molecular system with a very strong Os-N triple bond which is converted into an Os-NP species by P transfer. This is followed by reactions with sulfur, -NPCI, azides ... This is beautiful molecular chemistry, very elaborate and difficult synthetic chemistry, but I don't agree at all with the hook of the paper in the headline, introduction, throughout the manuscript: "PN" in the universe/space. Furthermore, the true novelty value is limited in my opinion.

We do thank Reviewer #3 for describing our work as "beautiful molecular chemistry". However, based on the subsequent critique brought forward by Reviewer #3, we, respectfully, disagree with this reviewer's evaluation.

First and foremost, in the introduction, we mention the occurrence of $P\equiv N$ in the interstellar medium, simply because this is important background information. We would like to bring to Reviewer #3's attention that the majority of previous investigations showcased in Chart 1 (Bertrand, Cummins, Schulz, and Smith) use the same parlance, which has effectively become agreed-upon standard. As such, the word "interstellar" occurs throughout these papers on $P\equiv N$ derivatives, so we see no reason why we should not use the same word for contextualization. Furthermore, it would be wrong, we believe, not to make an attribution to the work of the scientists that have established the existence of this rare molecule in the interstellar medium.

First things first:

(1) The authors have prepared an $Os=N=P$ complex starting from an osmium-nitrogen complex with an Os-N triple bond by P-transfer. Unfortunately, this has nothing to do with the isolated PN molecule in the gas phase / space. Niecke's Mes^*NP^+ is closer to "PN" -> true triple bond, except that a Niecke would never have thought of selling it that way. Sorry for these harsh words, but I am tired of the increasing overselling these days. At no point in the present synthesis paper can the authors show that they have even remotely actually generated a PN triple bond.

In this manuscript, we clearly distinguish between our cumulenic $[Os^{IV}=N=P]$ complex and the isolated $P\equiv N$ molecule in the gas phase/interstellar medium. We do not claim any direct generation of free $P\equiv N$ but rather

emphasize how elusive phosphorus-nitrogen bonding motifs can be stabilized through coordination to osmium. While Niecke's $\text{Mes}^*\text{N}\equiv\text{P}^+$ complex indeed exhibits a true $\text{P}\equiv\text{N}$ triple bond, our work explores a $\text{P}\equiv\text{N}$ -containing species that is stabilized by a metal and undergoes subsequent controlled transformations into useful species.

However, we do not agree with the postulate that $\text{P}\equiv\text{N}$ in the gas phase has nothing to do with $\text{P}\equiv\text{N}$ as a ligand. It is well established that gaseous molecules such as N_2 and CO may coordinate to metal complexes in solution; Cummins even demonstrated how $\text{P}\equiv\text{N}$ can be transferred to an iron complex, in solution. The overwhelming consensus among inorganic chemists is plainly to describe N_2 , CO , and PN complexes as containing neutral diatomic ligands and naming these after the gaseous diatomics (see papers by Smith and Cummins). We do not find it appropriate to confuse the scientific community by inventing an alternative naming system.

As for the remark about any "overselling", this is always a fine balance. But please note that Reviewer #2 specifically requests that we do a better job at evaluating "*the implications of such synthetic breakthrough*" – if anything, Reviewer #2 asks for more selling.

Finally, we do not think it is reasonable to engage in conjecture about what Prof. Niecke might have thought about the present study. Prof. Niecke's legacy in main-group chemistry is undeniable, and in acknowledgment, we have already placed the $[\text{Mes}^*\text{N}\equiv\text{P}]^+$ molecule as the first entry (**A**) of the first chart of this manuscript.

(2) P transfer via OCP^- has been known for a long time (see papers by Grützmacher, Goicoechea, Bertrand ...) - so nothing new either. The authors write: "Herein, we describe how sodium phosphoethynolate delivers a P atom to an electrophilic osmium nitride complex to form a neutral $[\text{OsN}\equiv\text{P}] \leftrightarrow [\text{Os}=\text{N}=\text{P}]$ motif." - without a reference to the work of the above authors. I know later there is one reference, but it should go here as well.

This comment, i.e., the precise location of a reference, we believe, is a matter of taste. We have already cited the work of said authors (as the reviewer acknowledges). These citations are even on the same page where the reviewer recommends that the new reference should go. We have pasted the references along with a few additional studies into the other paragraph.

In addition, we emphasize that our study extends beyond previously reported OCP^- reactivity by demonstrating its ability to generate a cumulenyl $[\text{Os}^{\text{IV}}=\text{N}=\text{P}]$ motif, which has not been previously described in this context.

(3) Transition metal complexes of the type M-NP are known, see work of Smith, Cummins and others.

We are fully aware, and agree, that M-NP complexes have been reported on two previous occasions. We present both of these earlier known M-NP complexes by Smith and Cummins in Chart 1 on the first page of the manuscript, so that this is clearly visible as the first display item that the reader sees. We fail to follow what the referee means by "others". We, through the present manuscript, are the only other group to report a M-NP complex, augmented with electronic structure studies and synthesis of exotic derivatives. Crucially, our study presents an electronically distinct $[\text{Os}^{\text{IV}}=\text{N}=\text{P}]$ species that exhibits novel reactivity, including transformations with sulfur, a chlorine atom transfer reagent, and cyclization with azide. These features differentiate our system from previously known M-NP complexes and provide new insights into P-N π -bonding and reactivity.

(4) R-N₄P heterocycles are known but are also not cited here and throughout the manuscript. If I remember correctly (Niecke?), these were also generated via a [2+3] cycloaddition reaction with azides. And of course, the Os system also has a Os-N bond, so it is simply wrong to write that an N₄P⁻ analogue is present.

Just as much as the C-H bond of methane differs from a metal-hydride bond, e.g. Os-H, the situation for an organic R-N₄P bond also fundamentally differs from that in Os-N₄P. Even though we draw a straight line to depict the bonds in these compounds, these bonds are fundamentally different in nature. In complex **6**, a [PN₄]⁻ ring is coordinated to osmium and should definitively be viewed as [PN₄]⁻ bound to Os^{III}. Accordingly, complex **6** is the first example of a tetrazaphospholide heterocycle without organic substituents.

Finally, in relation to the first point of the reviewer's criticism (lack of references to previous work), it is true that Niecke published an arylsubstituted derivative, Ar-N₄P, in 1993; furthermore, Schulz published a similar arylsubstituted heterocycle in 2006. In fact, we did cite these papers in the original manuscript, following the sentence "All other unsubstituted [P_nN_{5-n}]⁻ rings have remained unknown". In the revised manuscript, these references are now number 50 and 51, respectively. However, the reason that our study goes a step beyond this state of the art is that we assemble the [PN₄]⁻ ring without recourse to a sterically encumbering aryl substituent, therefore succeeding in the assembly of a conceptually simpler aromatic interpnictide system.

Taking these four points together, there is not much that is new in terms of new synthesis route, new molecular system, new insights into structure and bonding that would justify a publication in *Nature Communications*.

We would like to stress that our method of synthesis is new and that we have made new molecular systems with cumulenic PN bonding and reactivity, including oxidations using sulfur to form a trigonal planar phosphorus(V) derivative, which has no structurally characterized analog. While we acknowledge the reviewer's concerns, we are convinced that our work is an important contribution to phosphorus-nitrogen bonding and transition metal reactivity. We have refined our discussion for clarity (see points below) to ensure all relevant prior work is properly highlighted and further emphasized the important aspects of our study.

I also noticed a few other details that are a bit problematic:

5) For example, the authors write: " In theory, **3** can be described by three limiting resonance contributors, namely [Os^{II}] singly bonded to a neutral [P≡N] ligand, [Os^{IV}] doubly bonded to a dianionic [P=N]²⁻ ligand, or [Os^{VI}] triply bonded to a tetraanionic [P-N]⁴⁻ ligand. " And see also Figure 2: Jumps of +IV in the oxidation state within a resonance in the Lewis picture make no sense at all in my opinion.

The three drawings shown in Figure 2 are the classical resonance extremes that one needs to consider for characterizing the electronic structure of **3**, and the oxidation state 'jumps' are the classical 2, not 4. As we repeatedly articulate in the manuscript and fully and cooperatively supported by theory and various advanced spectroscopic means, the electronic structure of **3** is unambiguously characterized as [Os^{IV}=N=P].

However, we believe that the basis for the reviewer's concern is that oxidation states may be assigned in two alternative ways, namely by considering metal-ligand bonding as being ionic or covalent. One can use lithium fluoride as a simple model to illustrate the two approaches for assigning electrons to the metal and the ligand: When using an ionic oxidation state assignment, the valence electrons in LiF are assigned as belonging to F, which is the more electronegative element, and this results in a (Li⁺)(F⁻) description. When instead using a covalent

oxidation state assignment, the electrons in LiF can be viewed as belonging to a (covalent) Li–F bond, which is (conceptually) split in a homolytic fashion, and in this case, a (Li[•])(F[•]) description arises.

This poses a relevant question: Which method for assigning oxidation states reflects reality best? Reasonable arguments, based on experimental observations, can be made in favor of either method: In aqueous solution, LiF dissociates into hydrated Li⁺ and F⁻ ions, whereas in the gas phase, LiF dissociates into neutral Li and F atoms.

In the present paper, we simply choose one method for assigning oxidation states (ionic) and apply it consistently throughout the text. We prefer this particular choice because it conveys the notion of a metal ion (Mⁿ⁺) coordinated by main-group ligands having closed-shell configurations and obeying the octet rule. As such, the ligands can be imagined as having a separate existence as relatively stable ions/molecules. Moreover, when employing the ionic method for assigning oxidation states, the *d*-electron count can be directly assessed from the oxidation state of the metal, which allows for a straight-forward rationalization of spectroscopic and magnetic properties of most coordination complexes.

Finally, when considering a metal-ligand multiple bond through the lens of the ionic oxidation state assignment approach that we employ, the more electronegative element (N in our case) is assigned as having all electrons of the metal-ligand multiple bond in question. Thus, when we describe the possibility of the π -bonds as being localized to N \equiv P, this leads to an Os^{II} description, and at the other extreme, when we describe the π -bonds as being localized to Os \equiv N, an Os^{VI} description instead results.

It would be useful and informative if the authors would perform an NBO/NRT analysis and then find the appropriate theoretical basis for their Lewis notation. Then one should take a close look at the *d* orbital occupations at the Os, partial charges, etc., and can then find a proper theory-based description.

We have carried out the NBO analysis that was sought by Reviewer #3 for **3** (included in the SI). The results perfectly corroborate the conclusions of our MO analysis, bond order calculations and QTAIM studies provided in the original version of the manuscript. That is, we provided a “*proper theory-based description*” in the first place, and our original conclusions have not changed with the additional calculations.

Specifically, our NBO results confirm the cumulenenic structure of our [Os^{IV}=N=P] complex: It shows two low-occupancy lone pairs localized on the nitrogen, two Os–P ‘antibonding’ interactions that are significantly Os-based (70–75 %), the corresponding bonding combinations are also semi filled (0.32–0.33) and P-centered, a P–N σ -bond, and an Os–N σ -bond. Given the localized nature of NBOs, this is a convoluted description, but it characterizes an [Os=N=P] fragment. In view of this agreement, our NBO analysis accurately captures the electronic structure of **3** and supports our original resonance description. Details are in the SI **Tables S9.2.3** and **S9.2.4**. In any event, we would like to emphasize that Natural Bond Orbitals are localized few-center orbitals (few typically being 1 or 2), which are useful to describe Lewis-structures on optimal compact form. However, for the purpose of our study, a local framework is not the most ideal for describing highly delocalized systems such as [Os=N=P], [Os=N=PS₂], etc.. – simply because the explanation becomes convoluted. Moreover, NRT analysis is impractical for large molecules such as **3** (we model a non-truncated molecule with 108 atoms, and not a small idealized model), as NRT describes, for example, aromatic delocalizations with multiple resonance structures, and thus, for **3**, one would end up with hundreds of contributing Lewis structures.

Without further considerations on how local/delocalized theoretical descriptions converge, we would like to draw the reviewer's attention to the X-ray Absorption Near Edge Structure (XANES) spectroscopy studies in our work – with which we completely integrate experimental data with the quantum mechanics. With XANES spectroscopy, we have spectroscopically probed the core-to-valence band energies and, importantly, the 5d-orbital occupations of the osmium centers in **2**, **3**, **4**, and **5**. In other words, we offer an experimentally founded oxidation state assignment for our complexes, in perfect agreement with calculations.

6) Regarding the Mayer BO, I would also like to point out that BO in general are not physical observables and should therefore be discussed with extreme caution. If I remember correctly, Mulliken and Mayer's BOs are heavily dependent on the base sets, as Schleyer already noted in the 1990s. And diffuse functions should not be used to calculate Mayer BOs either. BOs therefore only make sense in comparison; the absolute values without comparison only provide limited information.

We do not use the Mayer BOs quantitatively but rather qualitatively to provide a general theoretical description of the bonding in our systems, and integrate this with our MO analysis and QTAIM studies. In support of our conclusions, we have also included Wiberg bond orders in the SI, which further corroborate our findings. These bond order analyses are also consistent with NBO analysis, MOs, and experiments, which further substantiate our methodologies and conclusions for the description of the studied unique bonds.

As far as the basis set goes, we use Def2-TZVP, which does not contain diffuse functions. It is a valence triple-zeta polarized basis set that, as repeatedly demonstrated, provides sufficient flexibility to describe complex wavefunctions.

About the 'extreme caution' when discussing computational results; we conducted our analyses according to the highest scientific standards, just as much as when we described Vanadium(II) Forming a Nitride (*J. Am. Chem. Soc.*, **2012**, *134*, 13035-13045), Titanium-Carbon Multiple Bonds (*J. Am. Chem. Soc.*, **2013**, *135*, 14754-14767), Cyclo-P₃ Complexes of Vanadium (*J. Am. Chem. Soc.*, **2015**, *137*, 15247-15261), An Iron(IV)-Nitride (*J. Am. Chem. Soc.*, **2017**, *139*, 15691-15700), Palladium-Palladium Transmetalation (*Nat. Commun.* **2018**, *9*, 4814), A Scandium-Diisoposphaethynolate (*Angew. Chem. Int. Ed.*, **2018**, *57*, 1049-1052), Zirconium Nitride and Imide (*J. Am. Chem. Soc.*, **2018**, *140*, 17399-17403), A Cobalt-Nitride (*J. Am. Chem. Soc.*, **2020**, *142*, 8233-8242), P-Atom Transfer to Titanium-Isocyanides (*Angew. Chem. Int. Ed.*, **2021**, *60*, 17595-17600), A Ti(IV)-Nitrido Group (*Angew. Chem. Int. Ed.*, **2024**, *63*, e202404601), amongst many more, but these 10 representative examples highlight our significant contributions to theoretical/computational analyses for describing exotic metal-ligand functionalities.

7) The authors write "All other unsubstituted [P_nN_{n-5}]⁻ rings have remained unknown." The authors compare their Os-N₄P with the series of anions P₅⁻...P_nN_m⁻, which in my opinion is not correct, as the Os-N bond should have a polar covalent character. So, it more like a R-N₄P (which is known). Moreover, they write: "The Os-N₄P bond distance is alike the other Os-N_{imine/DMAP} bonds of **6**, indicating coordination through a dative σ-bond." This is not correct. The distance does not necessarily say anything about the nature of the bond! It needs theoretical backup. So, what tells NBO analysis? Did the authors observe heterolytic Os-N bond cleavage? By the way, the Os-N distance was found to be 2.069(7) Å, which is almost the sum of covalent radii (2.00 Å). Besides, the ORTEPs of the N₄P and Os in Figure 3 look as if there is something wrong and ORTEPs without giving the temperature are meaningless.

We first discuss our synthesis of the Os–N₄P complex in relation to the reports of the discrete salts, [P₅][–] (Baudler), [N₅][–] (Lu), and [P₂N₃][–] (Cummins), because these studies are important in the context of the isolation of a [PN₄][–] ring coordinated as a ligand to a transition metal (our study). We think it is correct to mention these studies because they contextualize our findings relating to the synthesis of the [PN₄][–] interpnictide.

Furthermore, we would like to reiterate, the R–N₄P bond is “like” that in Os–N₄P, just as much as the methane CH₃–H bond is like an osmium hydride Os–H bond – the orbital interactions involving a group bound to osmium (5d) *vis-à-vis* carbon (2s/2p) are very different. Then again, the description that “*the Os–N bond should have a polar covalent character*” seems to miss a central word namely “dative”, which, within the field of organometallic/inorganic chemistry, makes our system a [PN₄][–] ring coordinated to Os(III). When depicting a bond as a straight line, it simply represents different underlying bonding interactions in organic chemistry and in organometallic chemistry.

As for the inquiry about an NBO analysis, we conducted this additional work, which also supports the description of Os–N₄P being a single dative σ -bond (see SI, **Tables S9.2.11** and **S9.2.12**), in agreement with our results from analysing MOs (QROs) and bond orders. Thus, undoubtedly, theory showcases σ -bond character for Os–N₄P and π -delocalization within the [PN₄][–] ring. We have incorporated a short comment about this into the manuscript, but none of our original conclusions have changed.

About the reactivity of the Os–N₄P bond, we observe no Os–N bond cleavage in **6** whatsoever (neither heterolytic nor homolytic). This is backed by our IR spectroscopic studies, detailed in responses to Reviewers #1 and #2.

Regarding the ORTEPs of **6**, it should be noticed that it is challenging to grow large crystals of this complex for X-ray crystallographic characterization, and that this challenge is naturally exacerbated by the thermal instability of the complex. Hence, diffraction from **6** was generally weak compared to the other compounds in our study. As a result, the model of **6** has slightly enlarged thermal ellipsoids. Nonetheless, our X-ray crystallographic study unequivocally establishes the chemical identity of **6**.

For the broader discussion of whether an ORTEP plot is “meaningless”, it empirically turns out that Debye-Waller factors are sensitive to a wide range of factors. These relate not only to the crystal itself (e.g. mosaicity, disorder, thermal motion, molecular center of gravity), but also to the instrumental setup and to the data treatment (e.g. temperature, exposure time, absorption correction). We are fully aware that all experience shows how thermal ellipsoids become larger when the temperature increases, but practical experience also shows that so many other factors than temperature influence the precise value of the Debye-Waller factors. In essence, diffraction experiments conducted at a fixed temperature, and on the same compound, may very well show variations in the values of the thermal ellipsoids due to these other factors. For that reason, the exact “meaning” of a thermal ellipsoid plot is not straight-forward to assess.

Finally, we originally reported the measurement temperature in the crystallographic tables in the Supplementary Information (all diffraction studies were conducted at 100(2) K). As requested by the referee, we have now added the measurement temperature in the appropriate figure captions of the manuscript.

8) ESI

(salNdipp)₂(Cl)OsN]: only ¹H NMR data?

Compound **1** does not crystallize readily from common organic solvents, and thus, it is not straightforward to isolate pure samples of this compound. However, it is not a central aim of our study to isolate compound **1** (or

2). For that reason, only **3**, **4**, **5**, and **6** are included in the Methods section at the end of the article. As such, we have deliberately shown how **3** can be made in yields well above 90% starting from $(\text{Bu}_4\text{N})[\text{Os}(\text{N})\text{Cl}_4]$ and without the need to conduct a work-up of the intermediates **1** and **2**. We have simply structurally characterized osmium nitride complexes **1** and **2** to establish the pathway and intermediates leading to **3**. Nevertheless, in the revised manuscript, we have included full NMR spectroscopic characterization and elemental analysis data on **1** and **2**.

$[(\text{salNdipp})_2(\text{OTf})\text{OsN}]$: same?

For compound **2**, we reported full NMR spectroscopic characterization in the original manuscript. In the revised manuscript, we have complemented this with EA and stability studies, but again, we emphasize that complex **3** can be made directly from $(\text{Bu}_4\text{N})[\text{Os}(\text{N})\text{Cl}_4]$ without the need to conduct a work-up of **1** or **2**.

How can a yield be determined without knowing the composition of the bulk in the solid state?

We do agree that yield and solid-state composition go hand in hand. But, once again, conducting a work-up of compounds **1** and **2** is not needed to prepare the compound of interest (**3**) in high yield and excellent purity.

10) $[(\text{salNdipp})_2(\text{DMAP})\text{Os}(\text{N}_4\text{P})]$ (**6**) EA looks strange to me as the N value is rather high. Usually, azides tend to have a too small N value due to slow release of N_2 .

Even though we use an azide (Me_3SiN_3) to prepare compound **6**, this complex is not itself an azide, but instead a complex containing tetrazaphospholide, $[\text{PN}_4]^-$, as a coordinated ligand. As we note in the manuscript, compound **6** is susceptible to thermal decomposition, which occurs readily at room temperature; even crystalline samples stored at $-35\text{ }^\circ\text{C}$ in our glovebox undergo decomposition, which can be detected after a few days. In solution, the decomposition leads to $\text{P}\equiv\text{N}$ complex **3**, N_2 complex **7**, as well as another component rich in nitrogen and phosphorus. Then again, the decomposition products likely remain trapped in the crystalline lattices of the samples. Therefore, it is not unreasonable that the sample could retain the nitrogen and show a high content of this element. Nevertheless, we prepared new samples of **6** (in triplicate, see response to Reviewer #2) and submitted these for elemental analysis. The result that the measured nitrogen content is close to the calculated value is fully reproducible.

In summary, I would like to say that this manuscript is not a communication, but a full paper. The novelty value does not correspond to a communication. Therefore, I recommend rejecting the manuscript and sending it to a journal specializing in inorganic chemistry, e.g. *Inorg. Chem.* or *Dalton transaction* etc.

We regret that Reviewer #3 was not convinced of the novelty of our study, but we have refined our discussion to enhance the clarity of the manuscript and conducted the extra computational studies that were requested. As for the length of the manuscript, we refer to the content types considered by *Nature Communications*.

Reviewer #1 (Remarks to the Author):

The authors addressed my concerns, and the manuscript has been improved after making revisions according to the comments. Now, I recommend its publication on Nature Communications in its current form.

We are delighted that Reviewer #1 finds the current version of our manuscript suitable for publication in *Nature Communications*, and we appreciate the reviewer's constructive feedback, which elucidated important mechanistic scenarios for the $[\text{PN}_4]^-$ heterocycle.

Reviewer #2 (Remarks to the Author):

It is my opinion that the authors have now addressed the comments/suggestions/questions from all three reviewers. New data has been included to support and expand several of the points highlighted. Therefore, I am satisfied that the requirements have been fulfilled that the publication can now take place.

We are delighted that Reviewer #2 finds the current version of the manuscript suitable for publication in *Nature Communications*, and we enjoyed the multi-faceted feedback, which enhanced the clarity of our manuscript.

Reviewer #3 (Remarks to the Author):

The revision of the manuscript was carried out very carefully and in detail. My critical comments were also addressed in great detail. However, as far as my criticism of novelty is concerned, I cannot really see much that is new. Of course, the novelty value of a publication is not a physical quantity and is always subject to a subjective factor, but I stand by my criticism:

We are pleased to see that, while Reviewer #3 still has reservations about the novelty of our study, the reviewer, nonetheless, appreciates the enhanced clarity of our manuscript, which has resulted from the reviewing process. We appreciate how the revisions made in response to the detailed comments from Reviewer #3 have improved the scientific rigor throughout our manuscript. And finally, we are happy that the reviewer, ultimately, is of the opinion that our manuscript can be published in *Nature Communications*. Thank you!

- (i) PN / HPN known (by the way, ACIE e202414456 should be cited)

We agree that PN and the radical species HPN are known, but due to the orbital mismatch between the diffuse $3p$ orbitals of phosphorus and the more compact $2p$ orbitals of nitrogen, the $\text{P}\equiv\text{N}$ triple bond is very weak and prone to decompose at ambient temperature/pressure conditions. As such the chemistry of the free $\text{P}\equiv\text{N}$ molecule has been restricted to high-vacuum and/or cryogenic noble gas matrix isolation experiments (<10 K). In this context, one of the advances in our study is how coordination of the $\text{P}\equiv\text{N}$ ligand to an osmium center permits its reactivity to be studied under controlled conditions, allowing for isolation of a range of useful and hard-to-access main-group fragments. We fully agree that Zeng's recent matrix isolation study of PN / HPN is very interesting; this study was published close to when we first submitted our manuscript, and we have cited the study in the revised version of our manuscript.

- (ii) OCP known as P-transfer reagent

We do agree that there is precedent for using the phosphoethynolate ion as a P-atom transfer reagent. We have cited an additional amount of representative examples of P-atom transfer using $\text{Na}(\text{OCP})$, as per the recommendation of the reviewer, but we would like to stress that the main thesis of our manuscript revolves around chemistry of the $\text{P}\equiv\text{N}$ ligand.

(iii) MNP known (M = metal, and yes M = Os is new)

It is true that Smith and Cummins have isolated examples of P≡N as a ligand, coordinated to Fe and Mo, respectively. While the electronic structure of these systems can be well described as [M–N≡P], our osmium-based system is characterized by more extensive backbonding. This results in a distinct, cumulenenic [Os=N=P] electronic structure, and this is responsible for the singular oxidation chemistry of the complex with sulfur (two-fold S-atom transfer) and a chlorine source (bent NPCL geometry).

(iv) 2+3 cycloaddition of NP + N₃ fragment to give a N₄P heterocycle known

We do agree that the transformation of an aryl derivative such as 2,4,6-^tBu₃C₆H₂–N≡P⁺ with azide has been previously shown to generate the corresponding aryl-substituted Ar–N₄P heterocycle. However, what surprised us in the present study, was how it was not possible to directly convert our neutral P≡N complex [(salNdipp)₂(DMAP)Os(NP)] (**3**) with azide. Instead, we needed to conduct the cyclization *via* an oxidized and paramagnetic intermediate, [(salNdipp)₂(DMAP)Os(NPCL)], to form an osmium-coordinated [PN₄][–] interpnictide.

(v) N₄P heterocycle motif known

The previous reports by Schulz and Niecke of aryl-substituted Ar–N₄P heterocycles relied on attaching sterically encumbering groups, such as the case of the 2,4,6-^tBu₃C₆H₂–N₄P systems. In our case, however, we stabilize the aromatic interpnictide without appending a sterically protecting organic group and instead showcase how the unsubstituted [PN₄][–] heterocycle may coordinate to a metal center as a discrete group. In addition, the coordination of [PN₄][–] to a paramagnetic Os^{III} center also renders our system electronically distinct from the previously reported aryl derivatives.

The authors rightly point out that an Os–N bond (in OsN₄P) is certainly to be considered somewhat differently than a C–N bond (in C–N₄P) and further assume that it is correct to equate this with N₄P^(–) and compare it with P₅^(–) and N₅^(–). Here I would like to disagree since both N₅^(–) and P₅^(–) have actually been synthesized almost naked, namely in the presence of a weakly coordinating cation! This is the reason, for example, why P₅^(–) has only now (2025!) been isolated as a “naked” anion for the first time by Müller in the solid state (see ACIE <https://doi.org/10.1002/anie.202505853>, which maybe should be cited and discussed). The authors then point out in their reply that it is a dative Os–N bond, if I understand them correctly. Again, I would disagree, as they show no experiment in which N₄P^(–) is actually released (e.g. in solution) or separated from osmium (unlike N₅^(–) and P₅^(–)).

We thank Reviewer #3 for bringing the very recently published (28 March 2025) structural characterization of [P₅][–] by Müller to our attention. We have incorporated this interesting reference into the manuscript alongside Baudler’s original studies of [P₅][–] in solution. Müller’s work showcases how a naked [P₅][–] anion may be coordinated to an Fe^{II} center to form a sandwich complex incorporating an η⁵-P₅[–] fragment, and in our view, this establishes a notional connection between the naked and the metal-coordinated [P₅][–] fragments. The microscopic reverse process – i.e. the release of a [P₅][–] ligand (or a [PN₄][–] ligand) from a metal center to form the corresponding naked ion – is an intriguing possibility. We have not yet achieved such a transformation with our osmium-based system, but this is an inspiring idea, that we will pursue in the future.

And even the IUPAC emphasizes: “The synonym 'dative bond' is obsolete. (The origin of the bonding electrons has by itself no bearing on the character of the bond formed. Thus, the formation of methyl chloride from a methyl cation and a chloride ion involves coordination; the resultant bond obviously differs in no way from the C–Cl bond in methyl chloride formed by any other path, e.g. by colligation of a methyl radical and a chlorine

atom.)” IUPAC Compendium of Chemical Terminology, 5th ed. International Union of Pure and Applied Chemistry; 2025. Online version 5.0.0, 2025. <https://doi.org/10.1351/goldbook.C01329>.

The assignment of bonding electrons within a molecule as belonging exclusively to any one of its constituent fragments is indeed a challenging task, and IUPAC’s example using methyl chloride nicely illustrates the problem. In fact, this example goes very well hand-in-hand with our previous discussion of the experimental outcome of dissociation of LiF in the gas-phase (homolytic) vs. aqueous solution (heterolytic). Given the inherent challenge of assigning electrons/oxidation states in a physically meaningful way, even for some of the simplest conceivable compounds, we have simply made a choice to use a particular terminology (ionic oxidation state assignment) to describe the complexes synthesized in our study, and applied this choice consistently throughout the manuscript.

So, for me, the C-N bond in R-N₄P is a polar covalent bond, as is the Os-N bond in Ligand-Os-N₄P, unless the authors show experimentally that they observe (almost) naked N₅P⁽⁻⁾ in solution or in the solid state. Neither has happened. The claim to have generated somewhat N₄P⁽⁻⁾ (in the manuscript: “definitively be viewed as [PN₄]⁻ bound to Os^{III}”) is wrong to my mind.

Along the lines of this discussion, we find the possibility of releasing the [PN₄]⁻ ligand from the osmium center intriguing, and we will pursue this idea in the future. We are still not certain of how to achieve the conversion in practice (especially in light of the instability of the Os-N₄P complex toward both reduction and thermal decomposition). But we thank Reviewer #3 for providing inspiration for future synthetic endeavors in our group.

Furthermore, from my point of view, the X-ray structure (OsN₄P) with a R_{int} value of 0.2489 (see IURC recommendation below) is unacceptable. This would not be possible in my working group. The fact that the R_{int} value is much too large and almost twice as large as the $wr2$ value indicates problems in the solution and should actually lead to warnings in the checkcif, which I unfortunately do not have now. I would not allow publication with such bad R -values for my people.

(According to IUCr: In the IUCr checkCIF procedure, the R_{int} value should usually be less than 0.12. Higher values may indicate issues with the data or the crystal.)

The relatively high value of R_{int} for the [PN₄]⁻ complex ties back to our previous discussion of the thermal instability of complex [(salNdipp)₂(DMAP)Os(N₄P)] (**6**). This instability, coupled with the particular nature of the kinetics of crystallization, prevented us from obtaining large and strongly diffracting crystals of **6**. In practice, this led to relatively weak diffraction spots recorded in our X-ray diffraction experiment, and as a result, the signal intensity did not exceed the noise by as large a margin as we could have hoped for. In such a situation, where the noise is significant compared to the signal intensity, the weak reflections will be affected adversely by noise, and hence a high value for R_{int} comes about. In principle, this issue could be alleviated by setting the resolution limit of the dataset at a lower threshold (at which the reflections would be more intense and therefore less affected by the noise). This again leads to a compromise between, one the one hand, excluding an entire resolution shell and gaining a lowered value of R_{int} , or, on the other hand, retaining the higher resolution data and suffering from a higher value of R_{int} in the crystallographic studies. Overall, we set the resolution limits of our crystallographic data in such a way as to obtain a reasonable compromise between R_{int} and resolution. In specific relation to warnings evaluating R_{int} in the checkcif report, complex **6** displays a B-level alert stating that R_{int} exceeds 0.18 alongside a G-level alert stating that R_{int} exceeds 0.12.

In addition to the above-mentioned points, which led to my conclusion that I am not in favor of an acceptance of this manuscript in *Nature Comm*, there are also things that I still see differently, but I am of the opinion that certain exp. and theor. data can and may well be discussed differently. For example, the oxidation states, where I still think that it is not reasonable to have a difference of 4 electrons Os(II) to Os(VI) in a resonance scheme. Or the comparison with PN in the universe. Even if other colleagues have done so, this is not an argument for me, nor is it the comparison with N₂ or CO, which are known to thermally dissociate easily, which was not shown for PN. I'm just old school and love down-to-earth introductions 😊.

We appreciate that Reviewer #3 acknowledges how the assignment of oxidation states and the inclusion of background information for contextualization reflects a certain level of subjective decisions on the part of the authors. Along the lines of the comment of the possibility to dissociate P≡N from our osmium system, we think this is an intriguing concept, and we hope to eventually observe (and subsequently report) it experimentally.

To be fair, since two out of three reviewers voted very enthusiastically for this manuscript and I myself as an author have always been very happy when the editor then decided in my favor, I would like to encourage the editor to decide in favor of acceptance here as well, even if I see it differently (see my points (i)-(v)). Why? It is still beautiful chemistry, and the excellent molecular chemistry described here was carried out competently, and I greatly appreciate how difficult the experiments were to be conducted!

We sincerely thank Reviewer #3 for referring to our work as being excellent molecular chemistry. We hold Reviewer #3 in high regard for displaying the ability to consider the reservations in their own report against the backdrop of the more enthusiastic reports from the other reviewers, and ultimately hand down an evaluation in support of publication in *Nature Communications*.

The research groups of Reinholdt and Pederson describe a molecular system with a very strong Os-N triple bond which is converted into an Os-NP species by P transfer. This is followed by reactions with sulfur, -NPCl, azides ... This is beautiful molecular chemistry, very elaborate and difficult synthetic chemistry, but I don't agree at all with the hook of the paper in the headline, introduction, throughout the manuscript: "PN" in the universe/space. Furthermore, the true novelty value is limited in my opinion.

First things first:

(1) The authors have prepared an Os=N=P complex starting from an osmium-nitrogen complex with an Os-N triple bond by P-transfer. Unfortunately, this has nothing to do with the isolated PN molecule in the gas phase / space. Niecke's Mes*NP⁺ is closer to "PN" -> true triple bond, except that a Niecke would never have thought of selling it that way. Sorry for these harsh words, but I am tired of the increasing overselling these days. At no point in the present synthesis paper can the authors show that they have even remotely actually generated a PN triple bond.

(2) P transfer via OCP⁻ has been known for a long time (see papers by Grützmacher, Goicoechea, Bertrand ...) - so nothing new either. The authors write: "Herein, we describe how sodium phosphoethynolate delivers a P atom to an electrophilic osmium nitride complex to form a neutral [OsN≡P] ↔ [Os=N=P] motif. " - without a reference to the work of the above authors. I know later there is one reference, but it should go here as well.

(3) Transition metal complexes of the type M-NP are known, see work of Smith, Cummins and others.

(4) R-N₄P heterocycles are known but are also not cited here and throughout the manuscript. If I remember correctly (Niecke?), these were also generated via a [2+3] cycloaddition reaction with azides. And of course, the Os system also has a Os-N bond, so it is simply wrong to write that an N₄P⁻ analogue is present.

Taking these four points together, there is not much that is new in terms of new synthesis route, new molecular system, new insights into structure and bonding that would justify a publication in Nature Communications.

I also noticed a few other details that are a bit problematic:

5) For example, the authors write: " In theory, 3 can be described by three limiting resonance contributors, namely [Os^{II}] singly bonded to a neutral [P≡N] ligand, [Os^{IV}] doubly bonded to a dianionic [P=N]²⁻ ligand, or [Os^{VI}] triply bonded to a tetraanionic [P-N]⁴⁻ ligand. " And see also Figure 2:

Jumps of +IV in the oxidation state within a resonance in the Lewis picture make no sense at all in my opinion. It would be useful and informative if the authors would perform an NBO/NRT analysis and then find the appropriate theoretical basis for their Lewis notation. Then one should take a close look at the d orbital occupations at the Os, partial charges, etc. and can then find a proper theory-based description.

6) Regarding the Mayer BO, I would also like to point out that BO in general are not physical observables and should therefore be discussed with extreme caution. If I remember correctly, Mulliken and Mayer's BOs are heavily dependent on the base sets, as Schleyer already noted in the 1990s. And diffuse functions should not be used to calculate Mayer BOs either. BOs therefore only make sense in comparison; the absolute values without comparison only provide limited information.

7) The authors write "All other unsubstituted [P_nN_{n-5}]⁻ rings have remained unknown." The authors compare their Os-N₄P with the series of anions P₅⁻ ... P_nN_m⁻, which in my opinion is not correct, as the Os-N bond should have a polar covalent character. So, it more like a R-N₄P (which is known). Moreover, they write: "The Os-N₄P bond distance is alike the other Os-Nimine/DMAP bonds of 6, indicating coordination through a dative σ-bond." This is not correct. The distance does not necessarily say anything about the nature of the bond! It needs theoretical backup. So, what tells NBO analysis? Did the authors observe heterolytic Os-N bond cleavage? By the way, the Os-N distance was found to be 2.069(7) Å, which is almost the sum of covalent radii (2.00 Å). Besides, the ORTEPs of the N₄P and Os in Figure 3 look as if there is something wrong and ORTEPs without giving the temperature are meaningless.

8) ESI

(salNdipp)₂(Cl)OsN]: only ¹H NMR data?

[(salNdipp)₂(OTf)OsN]: same?

How can a yield be determined without knowing the composition of the bulk in the solid state?

10) $[(\text{salNdipp})_2(\text{DMAP})\text{Os}(\text{N}_4\text{P})]$ (6) EA looks strange to me as the N value is rather high. Usually, azides tend to have a too small N value due to slow release of N_2 .

In summary, I would like to say that this manuscript is not a communication, but a full paper. The novelty value does not correspond to a communication. Therefore, I recommend rejecting the manuscript and sending it to a journal specializing in inorganic chemistry, e.g. *Inorg. Chem.* or *Dalton transaction* etc.